# The impact of consumer preferences on the evolution of competition in China's automobile market under the Dual Credit Policy—A density game based perspective

**Ying Xie, Jie Wu ● *, Xiao Zhou, Yongxiang Sheng**

School of Economics and Management, Jiangsu University of Science and Technology, Zhenjiang, Jiangsu, China

* 182040043@stu.just.edu.cn

**Data Availability Statement:** All relevant data is contained within the article: The original contributions presented in the study are included in

## Abstract

The evolution of the automobile market is a macro-expression of the behavior of automakers' production decisions. This study examines the competitive environment between new energy vehicles (NEVs) and conventional fuel vehicles (CFVs) and develops a game-theoretical model incorporating consumer utility, automaker profit, and the competitive density of NEVs and CFVs. It aims to assess how consumers' preferences for vehicle range and smart features influence automakers' strategic decisions and the broader market evolution under the Dual Credit Policy. The findings indicate: (1) A low NEV credit price facilitates NEV market size growth, but this growth rate diminishes beyond a certain price threshold; (2) The lower the consumer's range preference, the higher NEV credit price can accelerate the development of new energy vehicles to their saturation value. However, when consumers in the market prioritize smart features, increasing the NEV credit price does not significantly influence the growth of NEV market size. (3) Higher consumer preferences for both range and smart features, combined with increased NEV credit prices, can synergistically accelerate the speed of the NEV market to reach the saturation value and also raise the saturation value of the scale of NEVs. And higher consumer range preference combined with increased NEV credit prices has a more significant effect on the promotion of NEV market size than the combined effect of higher consumer smart preference and increased NEV credit prices. The actual data of China's automobile market is used in the simulation of this model. The model and its simulation results effectively explain and reveal the evolutionary impacts of consumers' range and smart feature preference on the promotion of China's NEVs under the Dual Credit Policy to provide effective technological and theoretical support for the promotion of the sustainable development of China's NEV industry.

## Introduction

Accelerating the promotion of NEVs is essential for transforming the automobile industry and achieving the dual-carbon target. Recognizing this, the Chinese government has prioritized

the article/supplementary material, further inquiries can be directed to the corresponding author/s.

**Funding:** National Natural Science Foundation of China (72171122); General Program of Humanities and Social Science Research of the Ministry of Education (20YJA880044); General Program of Social Science Foundation of Jiangsu Province (22JYB014).

NEV promotion [1]. In 2017, the Ministry of Industry and Information Technology ("MIIT") issued the Measures for Parallel Management of Average Fuel Consumption and New Energy Vehicle Credit for Passenger Vehicle Enterprises ("the Dual Credit Policy"). The Dual Credit Policy prompts automakers to expand the business of NEVs through market-oriented means, quantifies the energy efficiency characteristics of automobiles using credit and sets target values, and has the policy guidance and penalty constraints of increasing the efficiency and limiting the production of conventional fuel automobiles, and increasing the production of NEVs and improving the efficiency of NEVs. China has become one of the world's most active markets for NEV sales. Yet, by the end of 2022, the country's NEV ownership amounted to 13.1 million vehicles, accounting for 4.1% of the total number of vehicles, and CFVs are still the mainstream of the consumer market. At the same time, with the ensuing radical automotive market transformation, the supply of credits in the NEV market exceeds the demand, and the price of credits fluctuates significantly, making the strategic significance of the Dual Credit Policy gradually fade out [2]. Therefore, whether the Dual Credit Policy can sustainably promote the promotion of NEVs and how to adjust the Dual Credit Policy according to the urgent Chinese auto market realities have become practical issues for the Chinese government to utilize institutional resources to guide the sustainable development of the NEV industry.

Studies have focused on the impact of the Dual Credit Policy on the automobile manufacturer's production and operation [3]. However, these studies overlook how consumer preferences can influence the effectiveness of such policies. Since consumers are the direct buyers of New Energy Vehicles (NEVs), their preferences directly affect behaviors, which, in turn, significantly influence the strategic decisions of automakers [4].

Smart products that leverage big data, artificial intelligence, and other digital technologies are increasingly becoming the consumer market's preference. Consequently, the intelligent user experience offered by these products, especially in the use of automobiles, has become a focal point. The primary competitive advantage of NEVs over CFVs lies in the smart experience they offer during operation [5]. This intelligent user experience and consumer preferences for smart features and performance are now key factors driving NEV adoption. A recent survey by J.D. Power shows that NEVs with high range have higher customer satisfaction, suggesting that consumers' range preference for NEVs is an important factor affecting the satisfaction and valuation of NEVs [6]. Although NEVs provide a smarter experience than CFVs, the "mileage anxiety" caused by low range remains a primary consumer concern [7]. Furthermore, the current research on the Dual Credit Policy and consumer preferences often lacks a micro-macro analytical perspective, rendering the results less practical for real-world automotive industry applications [8]. Therefore, considering both the Dual Credit Policy and consumer preferences, exploring the role of automakers' micro-production decisions and the macro-automobile market evolution mechanism is a necessary entry point for accelerating the promotion of NEVs.

The operational environment can influence the effectiveness of the Dual Credit Policy and consumer preferences in advancing NEVs. Research has focused on the interplay of cooperation and competition among automakers under the Dual Credit Policy [9, 10]. As NEVs are introduced into the market, they vie for the same limited resources as Conventional Fuel Vehicles (CFVs) within the market's environmental capacity. This cap on growth potential can lead to a competitive dynamic where the expansion of one category of vehicles constrains the other, thus influencing their respective market penetration rates [11, 12]. Employing a density game approach allows for an in-depth analysis of this competitive landscape and the evolutionary patterns of the market participants within the constraints of maximum market capacity. Utilizing the density game to study the competitive interplay between NEVs and CFVs offers

insights into how the Dual Credit Policy and consumer preferences tangibly affect the advancement of NEVs.

We constructed a competitive density game model of NEVs and CFVs to examine the dynamic evolution of production decision-making in NEV promotion. It investigates the impetus of market evolution from the perspectives of the Dual Credit Policy and consumer preferences and demands. This study aims to achieve several key objectives:

1. Establish a link between the micro-level production decisions of automakers and the larger evolution of the automobile market.

2. Investigate how the NEV credit price affects the expansion of the NEV market within a competitive landscape.

3. Examines the effects of consumer preferences—individual and combined preferences for vehicle range and technological features—on the evolution of the automobile market at the various NEV credit price levels.

The remainder of the paper is structured as follows: In Section 2, the relevant literature is reviewed. Section 3 introduces the methodology of this paper. Section 4 address the game results, proof, and analysis. Section 5 conducts numerical simulation. Section 6 summarizes.

## Literature review

The Dual Credit Policy has garnered substantial attention since its inception. In light of this, our study delves into the market's competitive dynamics and investigates the mechanism of consumer preference under the the Dual Credit Policy on the micro production decisions of automakers and the evolution of macro automobile market.

### The Dual Credit Policy

Since its implementation, the Dual Credit Policy has been a significant focus of scholarly concern, with current research primarily addressing macro and micro perspectives. At the macro level, Yangyang Jiao et al. [13] found that the Dual Credit Policy can replace financial subsidies to incentivize the promotion of NEVs. Yaoming Li et al. [14] found that the policy was an effective solution to expanding NEV's market share. Ou et al. [15] quantified and compared the impacts of the policy on the Chinese automobile market by using the NEOCC model, revealing that the industry finds it easier to meet the NEV ratio requirement rather than the CAFC target value under the present policy parameters. Wu et al. [16] analyzed the macro impact of the policy and noted that it requires more stringent CAFC points and the NEV credit system.

At the micro level, research on the policy has focused on using game theory and mixed-integer linear programming to explore automakers' production decisions. Li et al. [17] establish a game theory model to analyze the optimal production decisions for NEV manufacturers in a duo-oligopoly market, including CFV manufacturers capable of producing CFVs and NEVs under three typical channel strategies considering the Dual Credit Policy; Ma et al. [18] established a model of production decisions regarding technological innovations under the policy and demonstrated the positive incentives of the policy on technological innovations. Meanwhile, Lu et al. [19] found that the policy promoted the profit increase of NEV sales, but it harms social welfare. Furthermore, Cheng et al. [8] examined the production decision problem of CFV and NEV manufacturers under the policy, noting that a higher points price better supports promoting the expansion of NEVs compared with the setting of a proportion of high-energy vehicles. He et al. [20] used a dynamic programming model to study the optimal

timing for automakers to invest in electrification under the policy. This shows that the tightened policy may not necessarily encourage all automakers to invest in NEVs immediately. Furthermore, Ding et al. [21] used game theory to discuss the impact of the combination of the Dual Credit Policy and subsidy cancellation on the production decisions of automakers, and the study found that the policy alone has positive and negative impacts on automakers' production decisions.

In summary, the Dual Credit Policy helped promote NEVs and facilitate the development of NEVs. Still, studies at the macro level have not assessed the role of credit price on the automobile market. Moreover, micro-level studies are challenging to represent the dynamic decision-making process of automakers, and these studies have neglected the important impact of consumer preferences during policy implementation. Therefore, it is imperative to explore the micro-decision-making mechanisms of automakers and the resulting macro-automobile market evolution phenomena from consumer preferences on the demand side and credit prices on the supply side.

## Consumers' range and smart feature preferences for new energy vehicles

Consumer preferences differ in many ways, and numerous studies have considered the heterogeneity of consumer preferences [22, 23]. By analyzing consumer preferences for NEV attributes, Sungsoon Jang et al. found that consumers preferred long-range and autonomous driving performance [24]. Range, as one of the most important factors affecting the promotion of NEVs, has been an important factor influencing consumers' willingness to purchase [25, 26]. Franke et al. [27] showed that consumers' "range anxiety" is related to range preference; Xiong et al. [28] found that cab drivers have a stronger preference for battery capacity requirements, while private NEV consumers have no significant preference for long-range mileage with no considerable preference. Cecere et al. [29] suggest that automakers improve the quality of NEV batteries to increase range performance for greater NEV penetration. Yuanyuan Xu et al. [30] study the pricing decision problem in a two-tiered automotive supply chain by considering factors such as range and price. Therefore, consumers' range preference has become important when discussing automakers' production decision choices.

Recently, Saurabh Vaidya et al. [31] noted that the residents' consumption structure is constantly upgrading, i.e., Maslow's demand hierarchy is upgraded. Consumption upgrading changes the consumer's consumption concepts and preferences, and the sales of green and smart goods grow significantly. The new consumption era has come, and intelligent consumption is the future consumption trend [32]. NEVs have more intelligent components than CFVs, and intelligent performance is one of the advantages of NEVs over CFVs. Therefore, it is essential to assess consumer consumption's influence on automakers' behavioral decisions and to drive the promotion of NEVs with consumer intelligent features and range preferences.

Therefore, it is necessary to explore how the NEV credit price and consumer preferences affect the micro production decisions of automakers and the resulting macro automotive market evolution mechanism. The implementation environment often affects the synergistic effect of NEV credit price and consumer preferences. Some scholars have mainly explored the micro-level interactions among firms in the industry under the implementation of the Dual Credit Policy, such as studying the competition and cooperation between manufacturers of NEVs and CFVs under the Dual Credit Policy [33]. Considering the competition and substitution relationship between CFVs and NEVs, the growth of either party will further constrain the development of the other party. The development of the automobile industry is limited by the maximum market capacity [34], and the maximum market environment capacity determines the development potential of the automobile industry. It is more realistic to explore the

impact of NEV credit price and consumer preference on the macro automobile market evolution mechanism from the perspective of market competition.

## Methodology

### Model backgrounds

The " Dual Credit Policy" manages both "Corporate Average Fuel Consumption Credit" (i.e., CAFC credit) and "New Energy Vehicle Credit" (i.e., NEV credit). The purpose of the policy is to improve the production and sales of NEVs and promote energy saving and consumption reduction of CFVs, and to promote the coordinated development of the NEV industry to form a market-oriented mechanism. The research objects of this paper are new energy (NEV) passenger cars and conventional fuel (CFV) passenger cars. Under the Dual Credit Policy, CFVs with average fuel consumption value higher than the target value will generate negative CAFC credit, and lower than the target value can get positive CAFC credit; new energy vehicles with actual value higher than the target value can get positive credit. Automakers can offset this by developing and producing new energy vehicles on their own or by purchasing credits from other automakers.

### Density game

A density game is one of the methods to study the effect of the adaptive behavior of subjects within a population on the evolution of population size [11, 12]. Density game research population size development is not only affected by the adaptive behavior and competitive behavior of the subjects within the population but also affected by the environmental capacity of the population and links the adaptive behavior of the subjects within the population and the density limit of the population, which can be an excellent way to explore the characteristics of the different populations under the influence of the density limit, i.e., the influence of the maximum market capacity of the different populations' competitive characteristics and the evolution mechanism. Novak et al. [35] proposed the density game theory, and Huang et al. [36] introduced the theoretical framework of density game to establish a density game model to explore the role of inter-subjective game gains within a population in driving the development of population size. Density games have been applied to medical, social networks, and other research fields, such as Philip Gerlee et al. [37] based on the density game, studied the ecological interactions of two cell populations, tumor cytokine producers and tumor cytokine non-producers, on the evolution of tumor cell populations; Julian Barreiro-Gomez et al. [38] applied the density game to the time-varying information sharing network under the DMPC controller design research; Huang [39] based on the density game and quantum decision theory, to study the initial strategy state of the subjects in the population under the limited environmental capacity, the game gain and entanglement strength on the dynamics of the population. The original state of the model is shown in Eq (1).

$$
\begin{cases}
\dot{x} = r_i x_i \left(1 - \dfrac{x_T}{K_i}\right) i = 1, ..., n \\
K_i = \displaystyle\sum_{j=1}^{n} a_{ij} \dfrac{x_i}{x_T}.
\end{cases}
\tag{1}
$$

The original formula is shown in Eq (1), the $x_i$ of Eq (1) denotes the number of populations of different strategic selectors at a given time, $r_i$ denotes the net replication rate of different strategy selectors when not constrained by density, $K_i$ denotes the maximum environmental capacity, $a_{ij}$ denotes the payoff of strategy $i$ when strategy $i$ and strategy $j$ are played ($a_{ij} > 0$),

and $x_T = \sum_{i=1}^{n} x_i$ denotes the total number of all selector populations for strategy $i$. The model assumes that the population size of the system is influenced not only by the game payoff matrix but also by the different environmental capacities of the population, which links the game payoff matrix to the population density limit, indicating that individuals with higher payoffs are more powerful and can better utilize resources compared to their competitors.

## Model description and assumptions

(1) Hypothesis 1: Automaker: the main body of the dual-credit policy assessment for passenger car manufacturers. Considering the passenger car market, there are two types of automakers, respectively known as automaker 1 and automaker 2, both engaged in automobile production activities in the automobile market and production and sales balance. Automakers can only choose between producing new energy vehicles (NEVs) or producing conventional fuel vehicles (CFVs) in the market, and both of them have different maximum market capacities. Automakers can choose from the same set of actions (i.e., produce new energy vehicles, produce conventional fuel vehicles). Automaker 1 chooses to produce new energy vehicles at a rate of $m$ while choosing to produce conventional fuel vehicles at a rate of $1 - m$, and Automaker 2 chooses to produce new energy vehicles at a rate of $n$ while choosing to produce conventional fuel vehicles at a rate of $1 - n$. The selling prices of new energy vehicles and conventional fuel vehicles are $P_1$ and $P_2$, respectively. The production costs are $\kappa C$ and $C$, where $\kappa > 1$. The market demand for NEVs and CFVs, respectively is $D_1$ and $D_2$. The profits are $\pi_1$ and $\pi_2$.

(2) Hypothesis 2: Consumer: Assume that consumers can choose the set of actions as {buy new energy vehicle, buy conventional fuel vehicle, not buy}. Each consumer buys at most one vehicle, and the basic benefits (e.g., color, appearance, etc.) of NEVs and CFVs are equal. However, referring to the article [40], we assume that new energy vehicles are more environmentally friendly and less damaging to the environment. Consumer's willingness to pay is $v$, where $v \sim U[0,1]$, and due to the different environmental awareness of consumers, different valuations will be generated for new energy vehicles and conventional vehicles. The valuation of new energy vehicles and conventional vehicles are $v$ and $\theta v$, respectively. $\theta$ represents the discount factor of consumers' valuation of conventional vehicles. Referring to previous studies [30, 41], the technical performance of range and the technical performance of intelligence, affect consumer utility when buying a car. In terms of range, $l$ denotes the range performance of NEVs, and $\phi$ denotes consumers' preference for the range performance of new energy vehicles, i.e., the degree to which consumers recognize the range performance of new energy vehicles. NEVs have intelligent components that distinguish them from CFVs and provide a better driving experience compared to CFVs, assuming that $g$ denotes the level of intelligence of NEVs and $\rho$ denotes consumers' smart preference for new energy vehicles, i.e., the degree of consumers' recognition of the intelligent performance of new energy vehicles, $g \in [0,1]$, $\rho \in [0,1]$.

(3) Hypothesis 3: The Dual Credit Policy: Due to the assessment of the Dual Credit Policy, the NEV automaker earns NEV credits for each new energy vehicle produced $a$, and selling surplus NEV credits to CFV automakers can generate additional credit revenue $\mu_1 = apD_1$. To meet the assessment standards, CFV automakers must pay additional credit costs $\mu_2 = p_{b+\delta}D_2$, where $\delta$ is the NEV ratio requirement, and $b$ is the unit CAFC integral factor. The previous year's credit carryover is not considered.

(4) The symbols and meanings of the relevant parameters are shown in Table 1.

**Table 1. Symbols and their descriptions.**

| Parameters | | Parameters | |
|---|---|---|---|
| $P_1$ | Unit price of NEVs | $P_2$ | Unit price of CFVs |
| $\kappa$ | Cost factor per unit of NEVs versus per unit of CFVs | $C$ | Unit cost of CFVs produced |
| $D_1$ | Market demand for NEVs | $D_2$ | Market demand for CFVs |
| $v$ | Consumer valuation of NEVs | $\theta$ | Discount factor for consumer valuation of CFVs |
| $l$ | Range performance of NEVs | $\phi$ | Sensitivity coefficient of consumers to the range performance of NEVs |
| $\rho$ | Consumer Recognition of Intelligent Level of NEVs | $g$ | Intelligent level of NEVs |
| $\mu_1$ | Credit gains for automakers per NEV produced | $\mu_2$ | Cost of credits for automakers per CFV produced |
| $a$ | The average NEV credits | $p$ | The average market price of NEV credits |
| $b$ | The average CAFC credits | $\delta$ | Annual percentage target for NEV credits |
| $P_1$ | Unit price of NEVs | $P_2$ | Unit price of CFVs |

## Model construction

**Consumer utility function.**  Consumers make purchase decisions based on the utility of different products to maximize their utility. Referring to the studies of related scholars [42, 43], the utility functions of consumers under different decisions are as follows:

1. the utility of adopting NEVs $u_1$:

$$u_1 = v - p_1 + \phi l + \rho g \tag{2}$$

2. the utility of adopting CFVs $u_2$:

$$u_2 = \theta v - p_2 + l \tag{3}$$

3. the utility of not buying any cars $u_3$:

$$u_3 = 0 \tag{4}$$

When $u_1 > u_2$ and $u_1 > 0$, we can get $v > \frac{l - \rho g - \phi l + p_1 - p_2}{1 - \theta}$ and $v > p_1 - \phi l - \rho g$, so that $v_1 = \frac{l - \rho g - \phi l + p_1 - p_2}{1 - \theta}$, and when $v_1 < v < 1$, it is easy to know that the demand for NEVs $D_1 = \int_{v_1}^{1} f(v) dv = 1 - \frac{l - \rho g - \phi l + p_1 - p_2}{1 - \theta}$. When $u_1 < u_2$ and $u_2 > 0$, we can get $\frac{-l - \eta e_2 + p_2}{\theta} < v < \frac{l - \rho g - \phi l + p_1 - p_2}{1 - \theta}$, so that $v_2 = \frac{-l + p_2}{\theta}$. When $v_2 < v < v_1$, the demand for CFVs $D_2 = \int_{v_2}^{v_1} f(v) dv = \frac{(l - p_2) - \theta(\rho g + \phi l - p_1)}{\theta(1 - \theta)}$. When $u_2 < u_3$, we get $0 < v < \frac{-l + p_2}{\theta}$, consumers do not buy any cars.

Based on the heterogeneous choices of consumers with Eqs (1) (2) and (3), the market demand for NEVs and CFVs can be further derived, as shown in Eqs (5 and 6)

$$D_1 = \int_{v_1}^{1} f(v)dv = 1 - \frac{l - \rho g - \phi l + p_1 - p_2}{1 - \theta} \tag{5}$$

$$D_2 = \int_{v_2}^{v_1} f(v)dv = \frac{(l - p_2) - \theta(\rho g + \phi l - p_1)}{\theta(1 - \theta)} \tag{6}$$

**Automaker profit function.** Automakers make decisions accordingly based on the principle of profit maximization. The product sales profit of NEVs $\pi_1$ and that of CFVs $\pi_2$ by automakers are shown below.

$$\pi_1 = (p_1 - \kappa C)D_1 = (p_1 - \kappa C)\left(1 - \frac{l - \rho g - \phi l + p_1 - p_2}{1 - \theta}\right) \tag{7}$$

$$\pi_2 = (p_2 - C)D_2 = (p_2 - C)\frac{(l - p_2) - \theta(\rho g + \phi l - p_1)}{\theta(1 - \theta)} \tag{8}$$

Since $\frac{\partial^2 \pi_1}{p_1^2} < 0, \frac{\partial^2 \pi_2}{p_2^2} < 0$, it follows that $\pi_1$ and $\pi_2$ are convex functions with respect to $p_1$ and $p_2$ respectively. Therefore, let $\frac{\partial \pi_1}{p_1} = 0, \frac{\partial \pi_2}{p_2} = 0$, and solving by association yields

$$p_1 = \frac{C(2\kappa + 1) + (2 - \theta)(\rho g + \phi l) - l + 2(1 - \theta)}{4 - \theta} \tag{9}$$

$$p_2 = \frac{2C + \theta(\kappa C - \theta + 1 - \rho g - \phi l) + (2 - \theta)l}{4 - \theta} \tag{10}$$

Bringing $p_1$ and $p_2$ to $D_1$, $D_2$, $\pi_1$, and $\pi_2$ yields

$$D_1 = \frac{2 + C[1 + \kappa(\theta - 2)] + 2\rho g - \theta(2 + \rho g) - l[1 + (\theta - 2)\phi]}{(4 - \theta)(1 - \theta)} \tag{11}$$

$$D_2 = \frac{2l + \theta + C(\theta - 2 + \kappa\theta) - \theta(l + \theta + \rho g + \phi l)}{(4 - \theta)(1 - \theta)\theta} \tag{12}$$

$$\pi_1 = \frac{[l - C + (\theta - 2)(\rho g - \kappa C + \phi l) + 2(\theta - 1)]^2}{(4 - \theta)^2(1 - \theta)} \tag{13}$$

$$\pi_2 = \frac{[(\theta - 2)(l - C) + \theta(\theta - 1 + \rho g - \kappa C + \phi l)]^2}{(4 - \theta)^2(1 - \theta)\theta} \tag{14}$$

**Game modeling of production decisions of automakers.** Automakers under the Dual Credit Policy choose and adjust their decisions in conjunction with their corresponding fitness in the group. Combining the above assumptions yields the game payment matrix shown in Table 2. To facilitate the construction of the density game model for NEVs and CFVs, let $H_1 = \pi_1, H_2 = \pi_1 + \mu_1, H_3 = \pi_2 - \mu_2, H_4 = \pi_2$.

**Table 2. Game payment matrix for automakers.**

| automaker 1 | | automaker 2 | |
|---|---|---|---|
| | | **NEV($n$)** | **CFV($1 - n$)** |
| | **NEV ($m$)** | $H_1, H_1$ | $H_2, H_3$ |
| | **CFV ($1 - m$)** | $H_3, H_2$ | $H_4, H_4$ |

**Competitive density game model construction for NEVs and CFVs.** The density game model (as in Eq (15)) proposed by Novak [35] and Huang [36] is used to explore the competitive evolutionary behavior of CFVs and NEVs for the automobile market in the competitive density game model, as shown in Eq (15):

$$\begin{cases} \dfrac{dx(t)}{dt} = r_1 x \left( 1 - \dfrac{1}{H_1} \dfrac{x}{K_x} - \dfrac{1}{H_2} \dfrac{y}{K_y} \right) \\ \dfrac{dy(t)}{dt} = r_2 y \left( 1 - \dfrac{1}{H_4} \dfrac{y}{K_y} - \dfrac{1}{H_3} \dfrac{x}{K_x} \right) \end{cases} \tag{15}$$

In Eq (15), $x$, $y$ respectively denote the current market sizes of NEVs and CFVs in the auto market with both as a function of time ($x(t)$, $y(t)$).

$r_1$, $r_2$ represent the net replication rate of NEVs and CFVs in the vehicle market without environmental capacity constraints, respectively. Under the given conditions, both values are constant.

$K_x$, $K_y$ denote the maximum current environmental capacity in the auto market for the production of NEVs and CFVs, respectively.

$H_1$, $H_4$ denote the market revenue per unit in the automobile market where all automakers in the automobile market choose to produce NEVs, and all choose to produce CFVs.

$H_1 * K_x$ denotes the maximum market value added of NEVs in the auto Market in the current period.

$H_4 * K_y$ denotes the maximum market value added of CFVs in the auto market in the current period.

$H_2$, $H_3$ denote the coefficients of the role of CFVs on the unit market size of NEVs and the role of NEVs on the unit market size of CFVs when the two types of automakers choose to produce NEVs and CFVs to game each other, respectively.

$r_1 x$, $r_2 y$ denote the development trend of NEVs and CFVs in the automobile market, respectively.

$1 - 1/H_1 * K_x$, $1 - 1/H_4 * K_y$ denote the growth retardation coefficients of NEVs and CFVs in the automobile market due to the consumption of limited social resources, respectively.

From Eq (15) and Automakers under the Dual Credit Policy choose and adjust their decisions in conjunction with their corresponding fitness in the group. Combining the above assumptions yields the game payment matrix shown in Table 2. To facilitate the construction of the density game model for NEVs and CFVs, let $H_1 = \pi_1$, $H_2 = \pi_1 + \mu_1$, $H_3 = \pi_2 - \mu_2$, $H_4 = \pi_2$, it can be seen that Eq (15) describes the dual credit policy scenario, i.e., the competitive

density game model under the influence of market volume for NEVs or CFVs when automakers choose two different strategies for producing NEVs and CFVs.

## Model analysis

**Game model stability analysis.** The probability of automaker 1 producing an NEV is $m$, and the probability of producing a CFV is $1 - m$. The probability of automaker 2 producing a NEV is $n$, and the probability of producing a CFV is $1 - n$. $E_{11}$ is the expected revenue function of automaker 1 choosing to produce NEVs, $E_{12}$ is the expected revenue function of automaker 1 choosing to produce CFVs, $\bar{E}_1$ is the average expected revenue of automaker 1, $E_{21}$ is the expected revenue function of automaker 2 choosing to produce NEVs, $E_{22}$ is the expected revenue function of automaker 2 choosing to produce CFVs, and $\bar{E}_2$ is the average expected revenue of automaker 2.

$$\begin{cases} E_{11} = nH_1 + (1-n)H_2 \\ E_{12} = nH_3 + (1-n)H_4 \\ \bar{E}_1 = mE_{11} + (1-m)E_{12} \end{cases} \tag{16}$$

$$\begin{cases} E_{21} = mH_1 + (1-m)H_2 \\ E_{22} = mH_3 + (1-m)H_4 \\ \bar{E}_2 = nE_{21} + (1-n)E_{22} \end{cases} \tag{17}$$

Due to the limited rationality of the setup, automakers gradually adjust their strategies according to the external situation in the game process. For example, if the automaker's adaptation of producing "NEVs" is better than producing "CFVs", the number of automakers choosing this strategy increases afterward. The following is a replication of the dynamic equation to analyze the production strategy adopted by automakers, as shown in Eq (18).

$$\begin{cases} F(m) = \dfrac{dm}{dt} = m(E_{11} - \bar{E}_1) = m(m-1)[n(H_2 - H_4 + H_3 - H_1) - (H_2 - H_4)] \\ F(n) = \dfrac{dn}{dt} = n(E_{11} - \bar{E}_1) = n(n-1)[m(H_2 - H_4 + H_3 - H_1) - (H_2 - H_4)] \end{cases} \tag{18}$$

Let $F(m) = 0$, $F(n) = 0$; we can obtain 5 local equilibrium points: $O(0, 0)$, $A_1(0, 1)$, $B_1(1, 0)$, $C_1(1, 1)$, $D_1(m^*, n^*)$, $m^* = \frac{H_4 - H_2}{H_1 - H_2 - H_3 + H_4}$, $n^* = \frac{H_4 - H_2}{H_1 - H_2 - H_3 + H_4}$.

According to Friedman's method [44] the stability analysis of the Jacobi matrix of the above equations leads to the system evolutionary stability strategy (ESS). From the system of equations in Eq (19), the Jacobi matrix $J_1$ can be calculated as:

$$J_1 = \begin{pmatrix} (1-2m)[H_2 - H_4 + n(H_1 - H_2 - H_3 + H_4)] & m(1-m)(H_1 - H_2 - H_3 + H_4) \\ n(1-n)(H_1 - H_2 - H_3 + H_4) & (1-2n)[H_2 - H_4 + m(H_1 - H_2 - H_3 + H_4)] \end{pmatrix} \tag{19}$$

The stability of the equilibrium point of the system of Eq (18) can be determined by the sign of the determinant $Det\,J_1 > 0$ and the trace $TrJ_1 > 0$ of the matrix $J_1$, as shown in Eq (19). When $Det\,J_1 > 0$ and $TrJ_1 > 0$ are satisfied, the equilibrium point of the replicated dynamic equation is the evolutionary stabilization strategy (ESS). Therefore, $Det\,J_1$ and $TrJ_1$ of Eq (20)

**Table 3. Determinants and traces of the evolutionary equilibrium point of the game model.**

| Equilibrium Point | Det $J_1$ | Tr$J_1$ |
|---|---|---|
| O(0, 0) | $(H_2 - H_4)^2$ | $2(H_2 - H_4)$ |
| $A_1(0, 1)$ | $(H_4 - H_2)(H_1 - H_3)$ | $H_1 - H_3 - H_2 + H_4$ |
| $B_1(1, 0)$ | $(H_4 - H_2)(H_1 - H_3)$ | $H_1 - H_3 - H_2 + H_4$ |
| $C_1(1, 1)$ | $(H_3 - H_1)^2$ | $-2(H_1 - H_3))$ |
| $D_1\left(\frac{H_4-H_2}{H_1-H_2-H_3+H_4}, \frac{H_4-H_2}{H_1-H_2-H_3+H_4}\right)$ | $\frac{(H_3-H_1)^2(H_2-H_4)^2(H_2-H_4+H_3)}{(H_1-H_2-H_3+H_4)^3}$ | 0 |

can be obtained as shown in Table 3.

$$\begin{cases} DetJ_1 = (1-2m)[H_2 - H_4 + n(H_1 - H_2 - H_3 + H_4)]*(1-2n)[H_2 - H_4 + m(H_1 - H_2 - H_3 + H_4)] - m(1-m)(H_1 - H_2 - H_3 + H_4)*n(1-n)(H_1 - H_2 - H_3 + H_4) \\ TrJ_1 = (1-2m)[H_2 - H_4 + n(H_1 - H_2 - H_3 + H_4)] + (1-2n)[H_2 - H_4 + m(H_1 - H_2 - H_3 + H_4)] \end{cases} \quad (20)$$

An analysis of the various stabilization points allows conclusions to be drawn:

Scenario 1: When $H_4 > H_2$, the evolutionary game system eventually converges to the point O (0, 0), and the combination of automakers strategies is (CFV, CFV). At the moment, two types of automakers in the market may choose to produce " CFVs "

Scenario 2: When $H_1 - H_3 - H_2 + H_4 < 0$, the evolutionary game system finally converges to the points $A_1(0, 1)$ and $B_1(1, 0)$, and the combination of automakers' strategies is (CFV, NEV) or (NEV, CFV). At this moment, two types of automakers in the market may choose to produce " CFVs" or " NEVs".

Scenario 3: When $H_1 > H_3$, the evolutionary game system finally converges to the point $C_1$ (1, 1), and the strategy combination is (NEV, NEV). At the moment, both types of automakers in the market choose to produce " NEVs".

Trend analysis of the evolution of strategy combinations of both sides of the game. Based on the above analysis, it can be seen that the strategies of automakers are different in different scenarios, which in reality is reflected in the stage of development changes. In scenario 1, the strategy of automaker 1 and automaker 2 is (CFV, CFV); and in scenario 2, the strategy of automaker 1 and automaker 2 is (CFV, NEV) or (NEV, CFV). In scenario 3 the strategies of automaker 1 and automaker 2 are (NEV, NEV), it can be seen that the evolutionary path of the whole evolutionary system is $(0,0) \Rightarrow (0,1)(1,0) \Rightarrow (1,1)$. As shown in Fig 1 the strategy combination of (CFV, CFV) as the initial stage of the strategic choices of automakers, the evolutionary stage of the strategic choices of automakers of (CFV, NEV) or (NEV, CFV), and the final shift to the ideal stage of the strategic choices of automakers of (NEVs, NEVs). Only when the difference between the profit of producing NEVs and the profit of producing CFVs minus the cost of points purchased to fulfill the assessment is large enough, the whole system can reach the ideal stage of "lock-in" in the evolutionary process of the game.

From the stability analysis of the evolutionary game strategy, it can be seen that under the influence of the Dual Credit Policy, the evolutionary path of the decision-making of the automakers in the market regarding the type of automobiles to be produced mainly follows the process of the initial stage—evolutionary stage—ideal stage, and needs to be regulated by the Dual Credit Policy, in order to advance the system evolution. In the evolution process, there will be a section of the automakers that choose to produce NEVs, while the other section of automakers will continue to produce CFVs. These two types of automakers produce competition in the market. In order to study the specific impact of the Dual Credit Policy regulation

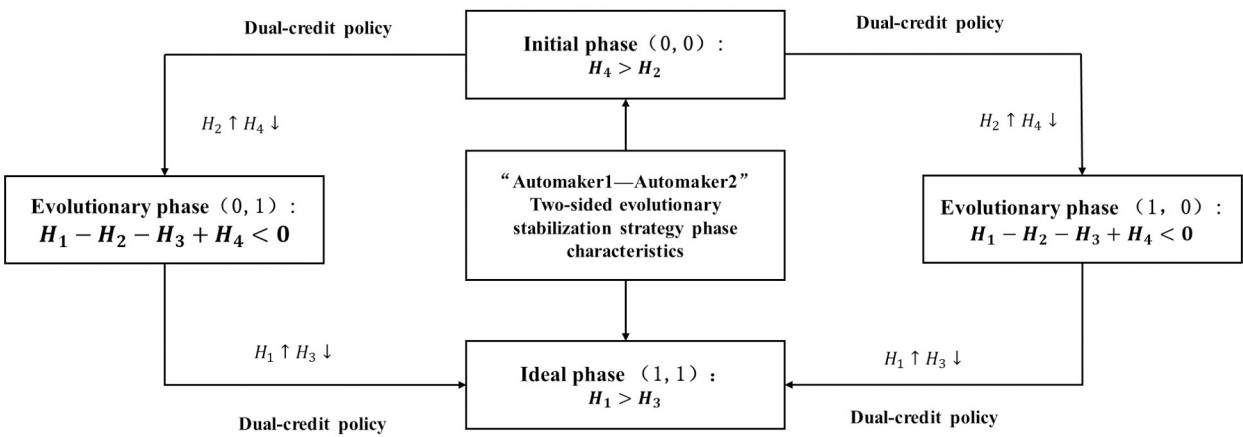

**Fig 1. Stages of the "Automaker1—Automaker2" two-party evolutionary stabilization strategy.**

means on the promotion of the production decision of the automakers, taking into account that the market is limited in resources, the automobile industry will be affected by the benefits of the market capacity constraints in the development of the automobile industry [34]; in fact, the party with higher benefits can more effectively utilize the ability of resources to achieve market diffusion, so further on in the competitive density game model for stability strategy solution.

## Competitive density game model stability analysis

**Competitive density game model stabilization point analysis.** In order to study the equilibrium state of the development process of NEVs and CFVs in the automobile market in the competitive density game model, so that $\frac{dx(t)}{dt} = \frac{dy(t)}{dt} = 0$, then there are four stabilization points in the system, i.e., $A_2(0,0)$, $B_2\left(0, H_4 K_y\right)$, $C_2(H_1 K_x, 0)$, $D_2\left(\frac{K_x H_1 H_3 (H_4 - H_2)}{H_4 H_1 - H_2 H_3}, \frac{K_y H_2 H_4 (H_1 - H_3)}{H_4 H_1 - H_2 H_3}\right)$.

The stability of the ordinary differential can be determined by the combination of the sign of the determinant and the sign of the trace of the Jacobi matrix of the system at the equilibrium point, then the Jacobi matrix can be derived from the set of differential equations as shown in Eq (21).

$$J_2 = \begin{bmatrix} r_1 - \dfrac{2r_1 x}{H_1 K_x} - \dfrac{r_1 y}{H_2 K_y} & -\dfrac{r_1 x}{H_2 K_y} \\[2ex] -\dfrac{r_2 y}{H_3 K_x} & r_2 - \dfrac{r_2 x}{H_3 K_x} - \dfrac{2r_2 y}{K_y H_4} \end{bmatrix} \quad (21)$$

From the stability theory of ordinary differential equations, the system must satisfy both *det* $J_2 > 0$ and *trJ*$_2 > 0$ at the equilibrium point. Combining the four local equilibrium points, the Jacobi matrix determinant and the trace of Jacobi matrix, the stability of the four equilibrium points can be obtained as shown in Table 4.

According to the summary, in the three stages of evolution paths of automaker 1 and automaker 2 in the market, when the profit of the automaker in producing NEVs plus its income from selling credits is less than its profit from producing CFVs, or less than its profit from producing CFVs after deducting the cost of purchasing credits, it is always necessary to regulate the Dual Credit Policy to promote the increase of the income from selling credits of the

**Table 4. Determinants and traces of evolving equilibrium points for competitive density game model.**

| Equilibrium Point | $detJ_2$ | $trJ_2$ | Stability conditions |
|---|---|---|---|
| $A_2$ (0,0) | $r_1 r_2$ | $-(r_1 + r_2)$ | - |
| $B_2$ (0, $H_4 K_y$) | $-r_1 r_2 (1 - H_4/H_2)$ | $r_2 - r_1 (1 - H_4/H_2)$ | $H_4 > H_2$ |
| $C_2$ ($H_1 Kx$, 0) | $r_1 r_2 (H_1/H_3 - 1)$ | $r_2 (H_1/H_3 - 1) + r_1$ | $H_1 > H_3$ |
| $D_2 \left( \frac{K_x H_1 H_3 (H_4 - H_2)}{H_4 H_1 - H_2 H_3}, \frac{K_y H_2 H_4 (H_1 - H_3)}{H_4 H_1 - H_2 H_3} \right)$ | $\frac{r_1 r_2 (H_2 - H_4)(H_3 - H_1)}{H_2 H_3 - H_1 H_4}$ | $\frac{r_2 H_2 (H_3 - H_1) + r_1 H_3 (H_2 - H_4)}{H_2 H_3 - H_1 H_4}$ | $H_1 - H_3 - H_2 + H_4 < 0$ and $sgn(H_2 H_3 - H_1 H_4) = +1$ |

automaker in producing NEVs, and to promote the increase of the cost of purchasing credits for producing CFVs to intervene in the development of NEV industry so as to obtain an evolutionary stabilization solution. The evolutionary stabilization solution is obtained by increasing the cost of purchasing credits of automakers producing CFVs and intervening in the development of the NEV industry. Matching the summarized stage evolution path with the equilibrium solution of the competitive density game model, the evolution path of the automobile market can also be analyzed.

Initial stage: when $H_4 > H_2$, the competitive evolution trend eventually converges to the equilibrium point $B_2$, at this time, CFVs' market share is high, for the original market share is very low for NEVs, it is difficult to enter the competitive evolution stage with them.

Evolutionary phase: when $H_1 - H_3 - H_2 + H_4 < 0$ and $sgn(H_1 H_4 - H_2 H_3) = +1$, the evolutionary trend of competition eventually converges to the equilibrium point $D_2$, at this time, a part of the automakers began to choose to produce NEVs, NEVs market share increases, the market share of CFVs shrinks, and the NEVs and CFVs competition is relatively moderate. The competition between NEVs and CFVs is relatively moderate, and each other can coexist in the market, and finally enter a state of equilibrium.

Ideal phase: When $H_1 > H_3$, the competitive evolution trend eventually tends to the equilibrium point $C_2$, at this time more automakers choose to produce NEVs, the market share of NEVs has increased significantly, the NEVs to the CFVs to bring a greater impact on the competition, which is far greater than the competition of the CFVs themselves, the CFVs will be eliminated in the market competition.

According to the statistics of China Association of Automobile Manufacturers (CAAM), the sales of NEVs in 2019, when the dual credit policy is formally implemented, amounted to 1.06 million units, with a market share of 4.99%, so this paper focuses on the evolution of competition between NEVs and CFVs under the evolutionary stage. When $H_1 - H_3 - H_2 + H_4 < 0$ and $sgn(H_1 H_4 - H_2 H_3) = +1$, the system is stable at the stabilization point $D_2$, i.e., both the NEVs and the CFVs finally reach the equilibrium state of competitive evolution.

**Competitive density game model stabilization strategy analysis.** From the analysis of stabilization points shown in Table 4, the conditions for the existence of stabilization points in the competing systems of NEVs and CFVs is $H_1 - H_3 - H_2 + H_4 < 0$ and $sgn(H_1 H_4 - H_2 H_3) = +1$. In this paper, we analyze the stability of the system by using the positional relationship of the isobars and the phase diagram analysis method combined with the track alignment, where the phase diagrams of isobars $l_1$ and $l_2$ (Fig 2) are shown below. Since $r_i$ and $x$, $y$ are greater than zero, the positive and negative of the system is determined by the equation in parentheses

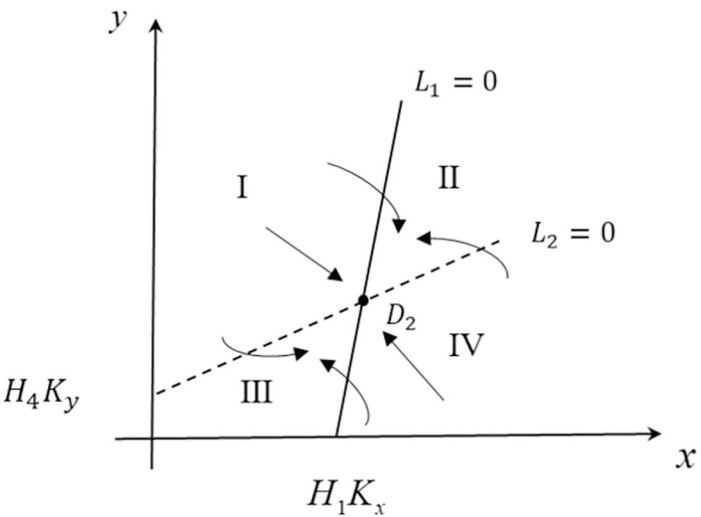

**Fig 2. Phase diagram of stabilization point $D_2$.**

in Eq (22).

$$
\begin{cases}
L_1 = 1 - \dfrac{1}{H_1}\dfrac{x}{K_x} - \dfrac{1}{H_2}\dfrac{y}{K_y} \Rightarrow
\begin{cases}
l_1 < 0 \Rightarrow dx/dt < 0 \\[2mm]
l_1 > 0 \Rightarrow dx/dt > 0
\end{cases} \\[6mm]
L_2 = 1 - \dfrac{1}{H_4}\dfrac{y}{K_y} - \dfrac{1}{H_3}\dfrac{x}{K_x} \Rightarrow
\begin{cases}
l_2 < 0 \Rightarrow dy/dt < 0 \\[2mm]
l_2 > 0 \Rightarrow dy/dt > 0
\end{cases}
\end{cases}
\tag{22}
$$

When satisfying $H_1 - H_3 - H_2 + H_4 < 0$ and $sgn(H_1 H_4 - H_2 H_3) = +1$, there exists positive slopes of isotropes $l_1$ and $l_2$, and the isotropes intersect in the first quadrant, where the slope of $l_1$ is greater than the slope of $l_2$. As can be seen from the phase diagram (Fig 2), the whole graph trajectory tends to the stabilization point $D_2\left(\frac{K_x H_1 H_3 (H_4 - H_2)}{H_4 H_1 - H_2 H_3}, \frac{K_y H_2 H_4 (H_1 - H_3)}{H_4 H_1 - H_2 H_3}\right)$, indicating that the point $D_2$ is the stabilizing equilibrium point. When the initial value is in I, the evolution characteristics of both sides are $dx/dt > 0$, $dy/dt < 0$, indicating that the growth rate of NEV market holdings is greater than zero and the growth rate of CFVs is less than zero; when the initial value is in II, the evolution characteristics of both sides are $dx/dt < 0$, $dy/dt < 0$, indicating that the growth rate of NEVs and CFVs are less than zero; when the initial value is in region III, the evolution characteristics of both sides are $dx/dt > 0$, $dy/dt > 0$, indicating that the growth rate of NEVs and CFVs are less than zero. When the initial value is in region III, the evolution of both sides of the game is characterized by $dx/dt > 0$, $dy/dt > 0$, indicating that the growth rate of both NEVs and CFVs is greater than zero; when the initial value is in region IV, the evolution of both sides of the game is characterized by $dx/dt < 0$, $dy/dt > 0$, indicating that the growth rate of NEVs is less than zero while the growth rate of CFVs is greater than zero in this region. growth rate are greater than zero.

## Scenario simulation analysis

In this study, based on the government and enterprise research data, related literature, and the basic assumptions of the model, the parameters related to the density game model are

parameterized, initial value assumptions are made, and the main parameter estimation and initial value settings are as follows.

## Dynamic evolution of the independent development of CFVs

Logistic equations are widely used in the field of population growth and market consumption demand forecasting. Without considering the competition of NEVs, the Dual Credit policy, and consumption preference, the long-term development trend of CFVs shows the characteristics of the logistic growth curve [45], and the corresponding Logistic equation is shown in Eq (23).

$$y(t) = \frac{K_y}{1 + \left(\frac{K_y}{K_1} - 1\right) e^{-r(t-t_0)}} \tag{23}$$

where $y$ denotes the market size of CFVs, $r$ denotes the natural growth rate, and $\_K_y$ denotes the maximum current market capacity of CFVs in the micro market.

Regarding parameter estimation, the parameter values in the Logistic equation are also solved using historical statistics. The Dual Credit Policy, which took effect in 2019, means that market sales volumes for New Energy Vehicles (NEVs) and Conventional Fuel Vehicles (CFVs) were not influenced by this policy before 2019. Consequently, the only reliable data samples for NEV market sales volume come from the past decade. Hence, this study uses the term 'maximum environmental capacity' for NEVs to refer to the market's capacity for CFVs, which is the benchmark in this context. Our analysis considers the market sales volume of conventional fuel passenger vehicles, excluding low-speed electric vehicles and agricultural vehicles, as our research sample. The study, therefore, uses data on CFV market sales from 1996 to 2018 to forecast the maximum potential market size for CFVs.

The research data on the sales volume of the CFV market from 1996 to 2018 are brought into Eq (7), which is fitted according to the research data using the nonlinear least squares method through MATLAB software. The fitting results are shown in Table 5 and Fig 3, and the parameter equations are in good agreement with the actual data. According to the fitting results of the table and figure as shown in Table 5, the market sales of CFVs in China around 2036 gradually reached saturation, with a saturation value of about 26,663,000 units. Therefore, this study sets the maximum capacity of the market of CFVs $K_y$ to 26.663 million units, and sets the maximum capacity of the market of NEVs $K_x$ to 30 million units because the market of NEVs is more robust in terms of sustainable development [45].

## Simulation background and initial value setting

Refer to Xuhui Wang et al. [46] for a twofold basis for the range of parameter values. One is the government and business research data and literature. Regarding policy research data, the Dual Credit Policy officially implemented in 2019, the "Chinese passenger car manufacturers average fuel consumption and NEV credit accounting table" and "China Automotive Industry Yearbook" set the initial value of a at 3.36 and of δ at 0.1. The average CAFC credits b initial value of 0.52. According to the NEV credits trading platform data, the NEV credit price in

**Table 5. Results of equation parameter estimation.**

| Parameter | Estimated Value | $R^2$ |
|-----------|-----------------|-------|
| $K_y$ | 2666.3 | 0.993 |

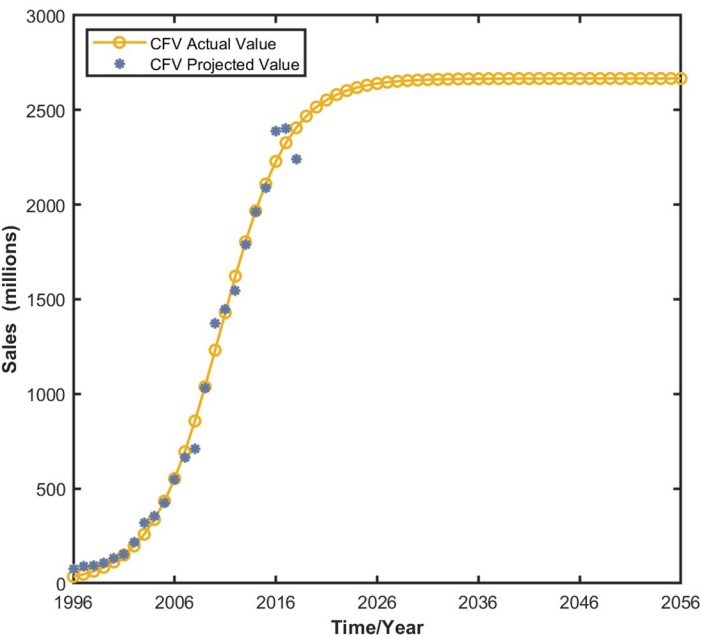

**Fig 3. Evolutionary trend of market share when CFVs are developed independently.**

2019–2022 is between 500–2088 yuan/points, and the initial value of the NEV credit price is set to be 0.05, p∈[0.05,0.2088]. In reference [20, 22], let $l = g = 0.4$, $\rho = \phi = 0.5$. In terms of corporate research data, a CFV cost is approximately 60% of the selling price [33]. The other is the principle of equilibrium. In the stability analysis of the density game model, to fulfill the realistic situation, it is necessary to satisfy that $H_1 - H_3 - H_2 + H_4 < 0$ and $sgn\,(H_1 H_4 - H_2 H_3) = +1$, and according to Eqs. (8)-(13), the calculations are set to $\kappa = 1.2$, $C = 0.1056$, $\theta \in [0,1]$.

Based on this, this paper uses the data of 2020 as a test, and the simulation results show that the simulation results of the NEV market sales and CFV market sales in 2020 are 1,171,651,000 and 2006,383,300, respectively, and their actual values are 1,246,000 and 1,868,0995,000, respectively, and the discrepancies between the simulation results and the actual data are -6.35% and 6.89% respectively, the data show that the error is less than 10% [47], so the behavior described by the model is basically consistent with the actual system behavior, which proves the validity of the model and can be used for further research. and the simulation results are basically consistent with the actual results.

## Impact of the Dual Credit Policy on the trend of automobile market evolution

Fig 4 shows the impact of NEV credit prices on the evolutionary trend of the automobile market. Keeping the other parameters of the market competition system for NEVs and CFVs unchanged, according to the Annual Report on the Implementation of Parallel Management of Average Fuel Consumption and New Energy Vehicle Credits for Passenger Vehicle Enterprises released by the Ministry of Industry and Information Technology, the average unit price of NEV credit price is CNY 500 per credit in 2019, CNY 1204 per credit in 2020, CNY 2088 per credit in 2021, and CNY 1,128 per credit in 2022. Therefore, set the price of NEV credits $p = 0.05$, $p = 0.1204$, $and\ p = 0.2088$.

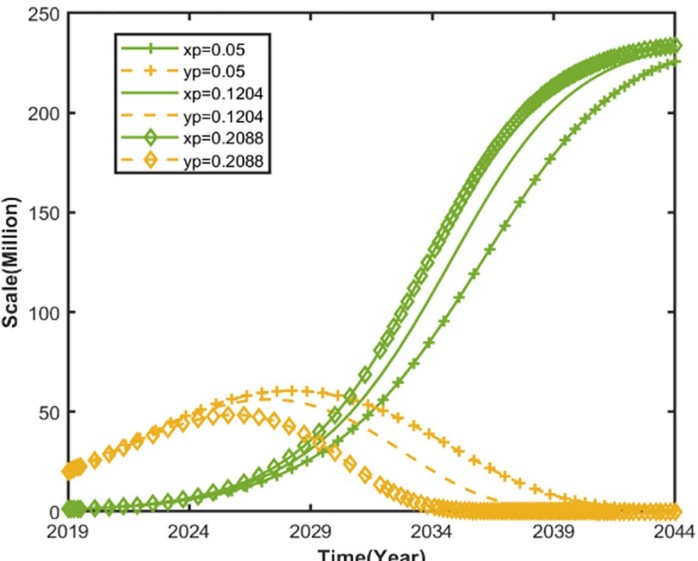

**Fig 4. Impact of NEV credit prices on evolutionary trends in the automobile market.**

The Dual Credit Policy coordinates the development of NEVs through market-oriented means, and the market price of NEV credit better reflects the market regulation effect of the Dual Credit Policy. Fig 4 shows that the competitive relationship between NEVs and CFVs gradually shows an increasing trend as the NEV credit prices increases. At different NEV credit prices, the market size of NEVs shows different growth trends. When the NEV credit price is less than CNY 1204 per credit, increasing it can accelerate the growth of NEV market size to its saturation value. However, it is difficult to increase the saturation value of NEV market size, and with the continuous growth of the credit price, this effect is weakened. Market size not only affects urban transportation development planning and government industrial policy making, but also reflects the development potential of the automobile market. Our findings also confirm those of Yitong Wang et al. [48]. A blind increase in the price of credits does not promote a sustainable increase in the size of the NEV market. When the price of NEV credits is too high, the government can act as the credits pool to improve the supply of credits when the credit supply exceeds the demand.

## Impact of consumer preferences on trends in automotive market evolution

As described by Shi [49] and Li [50], in order to achieve the sustained growth of new energy vehicle market size, policy intervention is necessary but not sufficient, because the market share growth of new energy vehicles is not only affected by macro policies, but also by consumer preferences, so in order to explore the impact of consumer preferences on the automobile market evolution, especially the NEV market evolution, under the Dual Credit Policy, we discuss how different consumer preferences affect the long-term evolution trend of the automobile market under different credit prices.

**Impact of consumer range preference on automotive market evolution trends.** Fig 5 shows the range preference of consumers for NEVs and the impact of the range performance of new energy vehicles on the evolutionary trend of the automobile market under different NEV credit prices, keeping the other parameters of the market competition system between

(a)

(b)

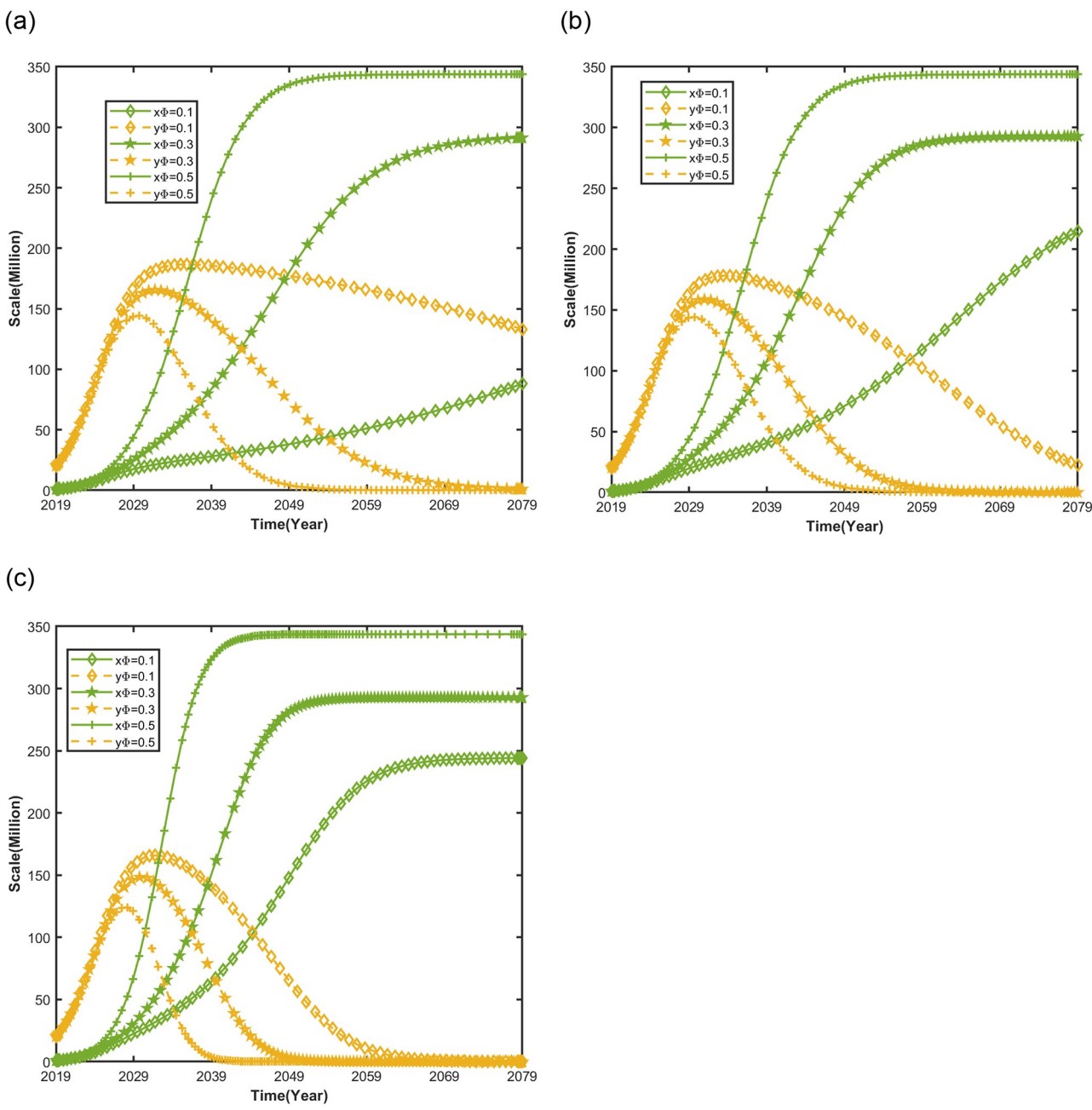

(c)

**Fig 5. Impact of consumer range preference on automotive market evolution trends.**

NEVs and CFVs unchanged, and setting NEV credit price $p$ = CNY 500 *per credit*, $p$ = CNY 1204 *per credit*, $p$ = CNY 2088 *per credit*, and consumers' range preference $\phi$ = 0.1, $\phi$ = 0.3, $\phi$ = 0.5.

As shown in Fig 5(a)–5(c), when the NEV credit price is unchanged, the increase in consumers' range preference can enhance the saturation value of the development of the new energy vehicle market. This is because the combined effect of consumer range preference and the formation of NEV credit price makes NEVs more competitive in the market, leading to a

competitive advantage to obtain the saturation value of a larger market size compared to competitors [51]. The lower the consumer's range preference, the higher NEV credit price can accelerate the development of new energy vehicles to their saturation value. Because, when consumer's range preference is low, the market acceptance of NEVs is low and the NEV credit price increases. This leads the government to utilize the credits deficit cost [52], which, in turn, prompts automakers to reduce the production of CFVs, focus on developing NEV range technology, subsidize the manufacturing cost of NEVs, and reduce the selling price of NEVs to increase consumer market demand for new energy vehicles.

**Impact of consumer smart preference on automotive market evolution trends.**   Fig 6 shows the impact of consumer smart preference on the evolutionary trend of the automobile market under different NEV credit prices, keeping other parameters of the competitive system of the market for NEVs and CFVs unchanged, and setting NEV credit price $p\phi CNY$ 500 *per credit*, $p\phi CNY$ 1204 *per credit*, $p\phi CNY$ 2088 *per credit*, and consumer smart preference $\rho = 0.1$, $\rho = 0.3$, $\rho = 0.5$.

As shown in Fig 6(a)–6(c), when consumers in the market prioritize smart features, increasing the NEV credit price does not significantly influence the growth of NEV market size. When consumers' smart preference is low, the increase in the NEV credit price is more significant enough to accelerate the development of NEVs to the saturation value of the market. Because,when consumers' smart preference of NEVs is low, the market acceptance of NEVs is low. This leads the government to utilize the credits deficit cost [53], which, in turn, prompts automakers to increase the production of NEVs, focus on developing smart technology for NEVs to increase consumer market demand for NEVs.

**Impact of dual consumer preferences on automobile market evolution trends.**   Fig 7 shows the impact of dual consumer preferences (range preference and smart preference) on the evolutionary trend of the automobile market under different NEV credit prices, keeping other parameters of the competitive system of the market for new energy vehicles and conventional vehicles constant, setting NEV credit price $p = CNY$ 500 *per credit*, $p = CNY$ 1204 *per credit*, $p = CNY$ 2088 *per credit*, consumers' range preference $\phi = 0.1$, $\phi = 0.3$, $\phi = 0.5$, consumers' smart preference $\rho = 0.1$, $\rho = 0.3$, $\rho = 0.5$.

As shown in Fig 7(a)–7(c), when consumers' preferences for range and smart of NEVs in the market is low, the competitive advantage of CFVs is greater than that of NEVs, and it is difficult to increase the competitive advantage of NEVs by increasing the NEV credit price. As consumers' preference for the range and smart of NEVs increase, the increase in the NEV credit price can increase the time for NEVs to replace CFVs, indicating that consumers' acceptance of NEVs is the prerequisite for their scalability [54]. Further, the Dual Credit Policy needs to be synergized with the consumers' willingness to buy NEVs. When consumers have the low smart preference and the high range preference for NEVs, compared to consumers have the high smart preference and the low range preference for NEVs, it is easier to increase the competitive advantage of NEVs, promote NEV market scale saturation value increase, and accelerate the time it takes NEVs to replace CFVs. It shows that consumers' higher range preference is more likely to increase the saturation value of NEV market size compared to consumers' improved smart preference. Accordingly, the higher range of NEVs is more likely to promote consumers' acceptance of NEVs than the smarter driving experience.

## Conclusion

In this study, we consider the competitive market environment and construct a competitive density game model of NEVs and CFVs considering the Dual Credit Policy and consumer preferences. Further, we explore the influence of consumers' range preference and smart

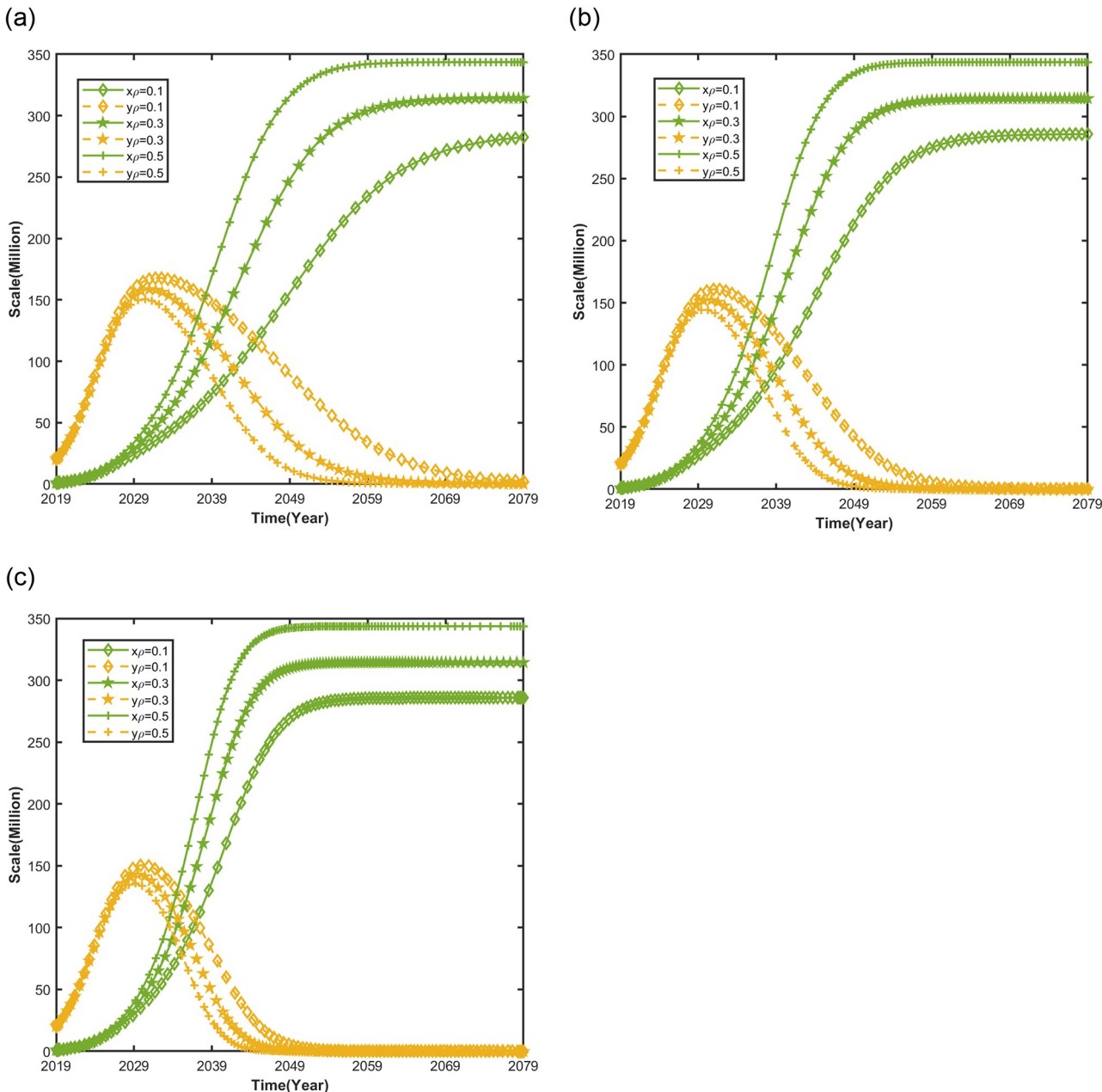

**Fig 6. Impact of consumer smart preference on automotive market evolution trends.**

preference on the micro-production decision of automakers and the long-term evolution of the macro-automobile market under the Dual Credit Policy.

1. A low NEV credit price facilitates NEV market size growth, but this growth rate diminishes beyond a certain price threshold. When the NEV credit price is less than CNY 1204 per credit, increasing it can accelerate the growth of NEV market size to its saturation valuem, but it is difficult to increase the saturation value of NEV market size, and with the continuous growth of the credit price, this effect is weakened.

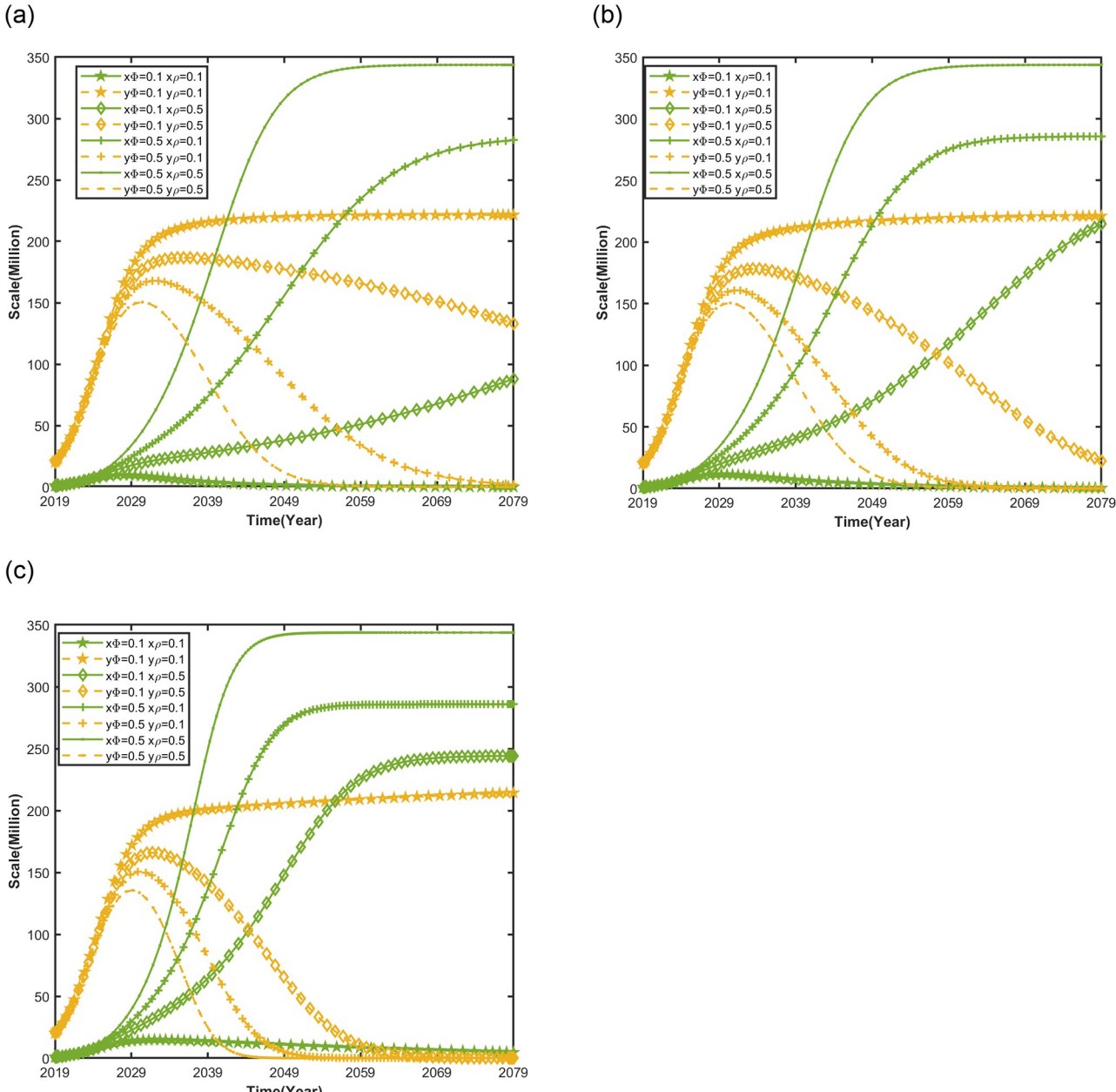

**Fig 7. Impact of dual consumer preferences (range and smart preference) on automobile market evolution trends.**

2. The lower the consumer's range preference, the higher NEV credit price can accelerate the development of new energy vehicles to their saturation value. When consumers' smart preference is low, the increase in the NEV credit price is more significant enough to accelerate the development of NEVs to the saturation value of the market.

3. Higher consumer preferences for both range and smart features, combined with increased NEV credit prices, can synergistically accelerate the speed of the NEV market to reach the saturation value and also raise the saturation value of the scale of NEVs. Higher consumer

range preference combined with increased NEV credit prices has a more significant effect on the promotion of NEV market size than the combined effect of higher consumer smart preference and increased NEV credit prices. When consumers have the low smart preference and the high range preference for NEVs, compared to consumers have the high smart preference and the low range preference for NEVs, it is easier to increase the competitive advantage of NEVs, promote NEV market scale saturation value increase, and accelerate the time it takes NEVs to replace CFVs.

The policy recommendations are as follows:

1. The optimization of the Dual Credit Policy is necessary to combine the demand-side support environment, adjust the Dual Credit Policy standards and the overall trading mechanism. Further, it is important to give full play to the role of policy combinations, such as planning and supporting the development route of NEVs, providing support for strengthening the core technology of NEVs, introducing purchase subsidies, tax exemptions, and other related policies and measures. This can be synergistic with the Dual Credit Policy to help automakers optimize according to consumer preferences, technical feasibility, and other available information. This can help make automakers make scientific technology choices, focus on key challenges such as power battery range experience and solving consumers' mileage anxiety,promote consumer market-oriented purchase choices, improve product technology level, and promote the benign cycle of NEV research and development (R&D), production, and consumption.

2. Automakers should take technological innovation as a key element to improve the performance and battery capacity of the entire vehicle, reduce production costs, increase the price advantage, increase investment in the field of intelligent driving and other advantageous areas of NEVs, reduce the gap with conventional vehicles, improve the technical performance of NEVs to leverage the improvement in marketing efforts.

Limitations and Prospects: 1) This paper only considers the decision-making problem of the two parties of automakers, and in the future, the government can be included in the scope of the game to study the optimal decision-making of the government based on the social welfare; 2) the data of the selected model may have a time lag;3)This paper investigates the evolutionary impact of three technological attributes of new energy vehicle range level, emission reduction level and intelligence level on the new energy vehicle market, and in the future, it will focus on the impact of these three types of technological attributes on the R&D decisions of new energy vehicle companies, their profits and the environment.

## Supporting information

**S1 File. Data.**
(DOCX)

**S2 File. Code.**
(ZIP)

**S3 File. Stability solutions for parameter changes in the game model.**
(DOCX)

## Acknowledgments

The authors deeply appreciate the valuable comments of anonymous reviewers and the editorial board for improving this study.

## Author Contributions

**Conceptualization:** Jie Wu.

**Data curation:** Xiao Zhou.

**Funding acquisition:** Yongxiang Sheng.

**Writing – original draft:** Ying Xie, Jie Wu.

**Writing – review & editing:** Ying Xie.

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
