## [Decision Letter · Decision Letter 0]

5 Oct 2023

PONE-D-23-29559The Impact of Dual Integral Policy and Consumer Preferences on the Evolution of Competition in China's Automobile Market - A Density Game Based PerspectivePLOS ONE

Dear Dr. wu,

Thank you for submitting your manuscript to PLOS ONE. After careful consideration, we feel that it has merit but does not fully meet PLOS ONE’s publication criteria as it currently stands. Therefore, we invite you to submit a revised version of the manuscript that addresses the points raised during the review process.

We look forward to receiving your revised manuscript.

Kind regards,

Grigorios L. Kyriakopoulos, 2 PhDs, 3 MSc, 2 MA, MEng, 2 BA, BSc

Academic Editor

PLOS ONE

Journal Requirements:

Reviewers' comments:

Reviewer's Responses to Questions

**Comments to the Author**

1. Is the manuscript technically sound, and do the data support the conclusions?

Reviewer #1: Partly

Reviewer #2: Yes

2. Has the statistical analysis been performed appropriately and rigorously? 

Reviewer #1: No

Reviewer #2: Yes

3. Have the authors made all data underlying the findings in their manuscript fully available?

Reviewer #1: Yes

Reviewer #2: Yes

4. Is the manuscript presented in an intelligible fashion and written in standard English?

Reviewer #1: No

Reviewer #2: Yes

5. Review Comments to the Author

Reviewer #1: The research article attempts to shed light on the complex dynamics surrounding the promotion of new energy vehicles (NEVs) in China. While the study delves into important aspects of the automobile industry and environmental sustainability, there are several critical issues that need addressing.

1. Lack of Clarity in Research Objectives: The article starts by emphasizing the significance of accelerating the promotion of NEVs but fails to provide a clear research question or objective. It mentions a broad array of topics, from government policies to consumer preferences, but does not specify a focused research problem. This lack of clarity makes it challenging for readers to understand the precise scope of the study.

2. Inadequate Literature Review: The literature review provided in the article is extensive but lacks critical analysis and synthesis. While it discusses prior research related to the Dual Credit Policy and consumer preferences, it fails to establish the gaps in existing literature that the current study aims to address. A more structured and critical examination of the relevant literature would have strengthened the foundation of the research.

3. Complexity of Model and Assumptions: The article introduces a complex model involving multiple automakers, consumer preferences, and the Dual Credit Policy. The assumptions made in the model are numerous and may oversimplify real-world dynamics. For instance, the assumption that consumers only consider environmental awareness as an additional utility for NEVs oversimplifies the myriad factors influencing consumer choices.

4. Lack of Empirical Data: The article relies heavily on modeling and assumptions without presenting empirical data or real-world evidence to support its claims. Empirical validation of the model's assumptions and predictions would have added credibility to the study.

5. Ambiguity in Findings: Despite the elaborate model and assumptions, the article's findings remain somewhat ambiguous. It discusses the impact of the Dual Credit Policy and consumer preferences on the NEV market but does not offer clear and actionable insights. The findings lack specificity and fail to provide guidance for policymakers or industry stakeholders.

6. Disconnected Sections: The article appears disjointed at times, with sections discussing various aspects of the research without clear connections. It would benefit from better organization and a logical flow of information to aid comprehension.

7. Language and Clarity: There are issues with language and clarity throughout the article. The writing can be convoluted and difficult to follow, making it challenging for readers to grasp the key points.

In conclusion, while the research article attempts to explore important aspects of the NEV market, it suffers from a lack of clarity in research objectives, an inadequate literature review, and the complexity of its model and assumptions. Additionally, the absence of empirical data and ambiguous findings weaken the overall contribution of the study to the field. To enhance its impact, the article should refine its research focus, provide a more critical literature review, simplify its model without oversimplifying reality, incorporate empirical evidence, and improve overall clarity in its presentation.

Reviewer #2: 1、The academic writing in English leaves much to be desired, as the manuscript contains a high number of grammatical errors. The manuscript needs to be checked by the author in detail.

2、The formatting of the manuscript is confusing, with many serious errors such as figures and figure names not appearing on a single page, missing text, large gaps, etc. For example, in line 518 of the manuscript, the content after ‘In’ is missing. In line 521 of the manuscript, the text after ‘Combined with’ is missing. The manuscript needs to be verified by the author in detail.

3、The title of the manuscript uses 'Dual Integral Policy'. But in the paper as well as in the abstract, the reference is to 'Dual Credit Policy'. There is a difference between the two statements. It is recommended that the author study it in detail and unify it into a more relevant one.

4、The manuscript is unclear in several places. For example, in line 183, the author mentions earlier that there are two automobile producers, each of which can produce two types of vehicles. Could the authors explain whether the intention here is to express that the basic benefits to consumers of the two types of vehicles produced by each producer (W1 or W2) want to be equal, or that the vehicles produced by both producers (W1 and W2) are equal?

5、In line 213 of the manuscript, the process of obtaining v1 through equations (1) and (2) represents the threshold of environmental awareness between no purchase behavior and the purchase of a NEV. Does it represent a problem with the conclusions derived from equations (1), (2), and (3), as well as the derivation of the later text, if it is written incorrectly here?

6、The manuscript proposes that ‘in the charging facilities in the construction of more backward areas, we should focus on the intelligent performance of new energy vehicles.’. Combined with real life, vehicle intelligence performance enhancement has little positive impact on consumers' choice of new energy vehicles if it is not convenient for consumers to charge.

7、The authors mainly consider the effects of changes in a single factor and do not consider the coupling between the factors, which is less relevant to the realities of life.

6. PLOS authors have the option to publish the peer review history of their article (what does this mean?). If published, this will include your full peer review and any attached files.

Reviewer #1: No

Reviewer #2: **Yes: **Zhang Shibo

---

## [Author Response · Author response to Decision Letter 0]

29 Nov 2023

Response to the Comments of Editors

Point 1: Please ensure that your manuscript meets PLOS ONE's style requirements, including those for file naming. The PLOS ONE style templates can be found at

Response 1: The formatting of the entire article has been updated to meet PLOS ONE’s style guidelines.

Point 2: In your Data Availability statement, you have not specified where the minimal data set underlying the results described in your manuscript can be found. PLOS defines a study's minimal data set as the underlying data used to reach the conclusions drawn in the manuscript and any additional data required to replicate the reported study findings in their entirety. All PLOS journals require that the minimal data set be made fully available. For more information about our data policy, please see http://journals.plos.org/plosone/s/data-availability.

Response 2: Data has been uploaded in accordance with plos one's data requirements.

Point 3: PLOS requires an ORCID iD for the corresponding author in Editorial Manager on papers submitted after December 6th, 2016. Please ensure that you have an ORCID iD and that it is validated in Editorial Manager. To do this, go to ‘Update my Information’ (in the upper left-hand corner of the main menu), and click on the Fetch/Validate link next to the ORCID field. This will take you to the ORCID site and allow you to create a new iD or authenticate a pre-existing iD in Editorial Manager. Please see the following video for instructions on linking an ORCID iD to your Editorial Manager account: https://www.youtube.com/watch?v=_xcclfuvtxQ

Response 3: My ORCID identifier is 0000-0002-3591-1141.

Response to the Comments of Reviewer 1

Thank you for your instructive and valuable comments and suggestions on our manuscript “The Impact of Consumer Preferences on the Evolution of Competition in China's Automobile Market under the Dual Credit Policy - A Density Game Based Perspective”（PONE-D-23-29559). We have studied all the comments, and have tried our best to revise the manuscript accordingly. The point-to-point responses to your comments are listed below. For ease of reading, the authors’ responses are listed in red. Changes are also shown in red in the revised manuscript.

Point 1: Lack of Clarity in Research Objectives: The article starts by emphasizing the significance of accelerating the promotion of NEVs but fails to provide a clear research question or objective. It mentions a broad array of topics, from government policies to consumer preferences, but does not specify a focused research problem. This lack of clarity makes it challenging for readers to understand the precise scope of the study.

Response 1: We thank you for your insightful comment, which raises both theoretical and practical considerations. We have rewritten the Introduction to clarity in research objectives. The details are as follows:

 Introduction

Accelerating the promotion of NEVs is essential for transforming the automobile industry and achieving the dual-carbon target. Recognizing this, the Chinese government has prioritized NEV promotion [1]. In 2017, the Ministry of Industry and Information Technology ("MIIT") issued the Measures for Parallel Management of Average Fuel Consumption and New Energy Vehicle Credit for Passenger Vehicle Enterprises ("the Dual Credit Policy"). The Dual Credit Policy prompts automobile manufacturers to expand the business of NEVs through market-oriented means, quantifies the energy efficiency characteristics of automobiles using credit and sets target values, and has the policy guidance and penalty constraints of increasing the efficiency and limiting the production of conventional fuel automobiles, and increasing the production of NEVs and improving the efficiency of NEVs. China has become one of the world's most active markets for NEV sales. Yet, by the end of 2022, the country's NEV ownership amounted to 13.1 million vehicles, accounting for 4.1% of the total number of vehicles, and CFVs are still the mainstream of the consumer market. At the same time, with the ensuing radical automotive market transformation, the supply of credits in the NEV market exceeds the demand, and the price of credits fluctuates significantly, making the strategic significance of the Dual Credit Policy gradually fade out[2]. Therefore, whether the Dual Credit Policy can sustainably promote the promotion of NEVs and how to adjust the Dual Credit Policy according to the urgent Chinese auto market realities have become practical issues for the Chinese government to utilize institutional resources to guide the sustainable development of the NEV industry.

Studies have focused on the impact of the Dual Credit Policy on the automobile manufacturer's production and operation[3]. However, these studies overlook how consumer preferences can influence the effectiveness of such policies. Since consumers are the direct buyers of New Energy Vehicles (NEVs), their preferences directly affect behaviors, which, in turn, significantly influence the strategic decisions of automakers [4].

Smart products that leverage big data, artificial intelligence, and other digital technologies are increasingly becoming the consumer market's preference. Consequently, the intelligent user experience offered by these products, especially in the use of automobiles, has become a focal point. The primary competitive advantage of NEVs over CFVs lies in the smart experience they offer during operation [5]. This intelligent user experience and consumer preferences for smart features and performance are now key factors driving NEV adoption. A recent survey by J.D. Power shows that NEVs with high range have higher customer satisfaction, suggesting that consumers' range preference for NEVs is an important factor affecting the satisfaction and valuation of NEVs[6]. Although NEVs provide a smarter experience than CFVs, the "mileage anxiety" caused by low range remains a primary consumer concern [7]. Furthermore, the current research on the Dual Credit Policy and consumer preferences often lacks a micro-macro analytical perspective, rendering the results less practical for real-world automotive industry applications [8]. Therefore, considering both the Dual Credit Policy and consumer preferences, exploring the role of automakers' micro-production decisions and the macro-automobile market evolution mechanism is a necessary entry point for accelerating the promotion of NEVs.

The operational environment can influence the effectiveness of the Dual Credit Policy and consumer preferences in advancing NEVs. Research has focused on the interplay of cooperation and competition among automakers under the Dual Credit Policy[9-10]. As NEVs are introduced into the market, they vie for the same limited resources as Conventional Fuel Vehicles (CFVs) within the market's environmental capacity. This cap on growth potential can lead to a competitive dynamic where the expansion of one category of vehicles constrains the other, thus influencing their respective market penetration rates [11-12]. Employing a density game approach allows for an in-depth analysis of this competitive landscape and the evolutionary patterns of the market participants within the constraints of maximum market capacity. Utilizing the density game to study the competitive interplay between NEVs and CFVs offers insights into how the Dual Credit Policy and consumer preferences tangibly affect the advancement of NEVs. 

We constructed a competitive density game model of NEVs and CFVs to examine the dynamic evolution of production decision-making in NEV promotion. It investigates the impetus of market evolution from the perspectives of the Dual Credit Policy and consumer preferences and demands. This study aims to achieve several key objectives: 

(1). Establish a link between the micro-level production decisions of automobile manufacturers and the larger evolution of the automobile market. 

(2). Investigate how the NEV credit price affects the expansion of the NEV market within a competitive landscape. 

(3). Examines the effects of consumer preferences—individual and combined preferences for vehicle range and technological features—on the evolution of the automobile market at the various NEV credit prices levels. 

Details are shown in the “Introduction “on line 40-153 of the revised version.

Point 2: Inadequate Literature Review: The literature review provided in the article is extensive but lacks critical analysis and synthesis. While it discusses prior research related to the Dual Credit Policy and consumer preferences, it fails to establish the gaps in existing literature that the current study aims to address. A more structured and critical examination of the relevant literature would have strengthened the foundation of the research.

Response 2: Thank you for your comment. We have rewritten the Literature Review. The details are as follows:

Literature review

The Dual Credit Policy has garnered substantial attention since its inception. In light of this, our study delves into the market's competitive dynamics and investigates the mechanism of consumer preference under the the Dual Credit Policy on the micro production decisions of automobile manufacturers and the evolution of macro automobile market.

The Dual Credit Policy 

Since its implementation, the Dual Credit Policy has been a significant focus of scholarly concern, with current research primarily addressing macro and micro perspectives. At the macro level, Yangyang Jiao et al. [15] found that the Dual Credit Policy can replace financial subsidies to incentivize the promotion of NEVs. Yaoming Li et al.[16]found that the policy was an effective solution to expanding NEV's market share. Ou et al. [17] quantified and compared the impacts of the policy on the Chinese automobile market by using the NEOCC model, revealing that the industry finds it easier to meet the NEV ratio requirement rather than the CAFC target value under the present policy parameters. Wu et al. [18] analyzed the macro impact of the policy and noted that it requires more stringent CAFC points and NEV points system.

At the micro level, research on the policy has focused on using game theory and mixed-integer linear programming to explore automakers' production decisions. Li et al.[19] establish a game theory model to analyze the optimal production decisions for NEV manufacturers in a duo-oligopoly market, including CFV manufacturers capable of producing CFVs and NEVs under three typical channel strategies considering the Dual Credit Policy; Ma et al.[20] established a model of production decisions regarding technological innovations under the policy and demonstrated the positive incentives of the policy on technological innovations. Meanwhile, Lu et al.[21] found that the policy promoted the profit increase of NEV sales, but it harms social welfare. Furthermore, Cheng et al.[8] examined the production decision problem of CFV and NEV manufacturers under the policy, noting that a higher points price better supports promoting the expansion of NEVs compared with the setting of a proportion of high-energy vehicles. He et al.[23] used a dynamic programming model to study the optimal timing for automakers to invest in electrification under the policy. This shows that the tightened policy may not necessarily encourage all automakers to invest in NEVs immediately. Furthermore, Ding et al.[24] used game theory to discuss the impact of the combination of the Dual Credit Policy and subsidy cancellation on the production decisions of automakers, and the study found that the policy alone has positive and negative impacts on automakers' production decisions.

In summary, the Dual Credit Policy helped promote NEVs and facilitate the development of NEVs. Still, studies at the macro level have not assessed the role of credit price on the automobile market. Moreover, micro-level studies are challenging to represent the dynamic decision-making process of automobile manufacturers, and these studies have neglected the important impact of consumer preferences during policy implementation. Therefore, it is imperative to explore the micro-decision-making mechanisms of automakers and the resulting macro-automobile market evolution phenomena from consumer preferences on the demand side and credit prices on the supply side.

Consumers' Range Preferences and Smart Preferences for New Energy Vehicles

Consumer preferences differ in many ways, and numerous studies have considered the heterogeneity of consumer preferences [25-26]. By analyzing consumer preferences for NEV attributes, Zhang et al. found that consumers preferred long-range and autonomous driving performance [27]. Range, as one of the most important factors affecting the promotion of NEVs, has been an important factor influencing consumers' willingness to purchase [28-29]. Franke et al. [30]showed that consumers' "range anxiety" is related to range preference; Xiong et al. [31] found that cab drivers have a stronger preference for battery capacity requirements, while private NEV consumers have no significant preference for long-range mileage with no considerable preference. Cecere et al.[32] suggest that automakers improve the quality of NEV batteries to increase range performance for greater NEV penetration. Yuanyuan Xu et al.[33] study the pricing decision problem in a two-tiered automotive supply chain by considering factors such as range and price. Therefore, consumers' range preference has become important when discussing automakers' production decision choices.

Recently, Saurabh Vaidya et al.[34] noted that the residents' consumption structure is constantly upgrading, i.e., Maslow's demand hierarchy is upgraded. Consumption upgrading changes the consumer's consumption concepts and preferences, and the sales of green and smart goods grow significantly. The new consumption era has come, and intelligent consumption is the future consumption trend [35]. NEVs have more intelligent components than CFVs, and intelligent performance is one of the advantages of NEVs over CFVs. Therefore, it is essential to assess consumer consumption's influence on automobile manufacturers' behavioral decisions and to drive the promotion of NEVs with consumer intelligent features and range preferences.

Therefore, it is necessary to explore how the NEV credit price and consumer preferences affect the micro production decisions of automakers and the resulting macro automotive market evolution mechanism. The implementation environment often affects the synergistic effect of NEV credit price and consumer preferences. Some scholars have mainly explored the micro-level interactions among firms in the industry under the implementation of the Dual Credit Policy, such as studying the competition and cooperation between manufacturers of NEVs and CFVs under the Dual Credit Policy[36]. Considering the competition and substitution relationship between CFVs and NEVs, the growth of either party will further constrain the development of the other party. The development of the automobile industry is limited by the maximum market capacity [37], and the maximum market environment capacity determines the development potential of the automobile industry. It is more realistic to explore the impact of NEV credit price and consumer preference on the macro automobile market evolution mechanism from the perspective of market competition.

 Details are shown in the “Literature Review “on line 154-328 of the revised version.

Point 3: Complexity of Model and Assumptions: The article introduces a complex model involving multiple automakers, consumer preferences, and the Dual Credit Policy. The assumptions made in the model are numerous and may oversimplify real-world dynamics. For instance, the assumption that consumers only consider environmental awareness as an additional utility for NEVs oversimplifies the myriad factors influencing consumer choices.

Response 3: Thank you for your comment. This is a very theoretical and practical question. The reviewer pointed out that the statement " the assumption that consumers only consider environmental awareness as an additional utility for NEVs " is an ambiguous statement in our article, and we have revised it in the second modeling assumption in the article.

Our article considered how the utility of consumers purchasing new energy vehicles is affected by the range and smart performance of the new energy vehicles. The article explores the mechanism of consumers' new energy vehicle range preference and smart preference on the evolution of automobile market under the Dual Credit Policy. This is because smart products that leverage big data, artificial intelligence, and other digital technologies are increasingly becoming the consumer market's preference. Consequently, the intelligent user experience offered by these products, especially in the use of automobiles, has become a focal point. The primary competitive advantage of NEVs over CFVs lies in the smart experience they offer during operation [5]. This intelligent user experience and consumer preferences for smart features and performance are now key factors driving NEV adoption. Besides, a recent survey by J.D. Power shows that NEVs with high range have higher customer satisfaction, suggesting that consumers' range preference for NEVs is an important factor affecting the satisfaction and valuation of NEVs[6]. Although NEVs provide a smarter experience than CFVs, the "mileage anxiety" caused by low range remains a primary consumer concern [7]. Sungsoon Jang et al. found that consumers preferred long-range and autonomous driving performance [24]. Range, as one of the most important factors affecting the promotion of NEVs, has been an important factor influencing consumers' willingness to purchase [25-26]. Franke et al. [27]showed that consumers' "range anxiety" is related to range preference; Xiong et al. [28] found that cab drivers have a stronger preference for battery capacity requirements, while private NEV consumers have no significant preference for long-range mileage with no considerable preference. Cecere et al.[29] suggest that automakers improve the quality of NEV batteries to increase range performance for greater NEV penetration. Yuanyuan Xu et al.[30] study the pricing decision problem in a two-tiered automotive supply chain by considering factors such as range and price. Therefore, consumers' range preference has become important when discussing automakers' production decision choices.

The modifications to the second modeling assumption in our article are as follows:

Hypothesis 2: Consumer: Assume that consumers can choose the set of actions as {buy new energy vehicle, buy conventional fuel vehicle, not buy}. Each consumer buys at most one vehicle, and the basic benefits (e.g., color, appearance, etc.) of new energy vehicles and conventional vehicles are equal. However, referring to the article [40] , we assume that new energy vehicles are more environmentally friendly and less damaging to the environment. Consumer's willingness to pay is ν, where v~U[0,1], and due to the different environmental awareness of consumers, different valuations will be generated for new energy vehicles and conventional vehicles. The valuation of new energy vehicles and conventional vehicles are ν and θν, respectively. θ represents the discount factor of consumers' valuation of conventional vehicles. Referring to previous studies [30,41], the technical performance of range and the technical performance of intelligence, affect consumer utility when buying a car. In terms of range, l denotes the range performance of NEVs, and ϕ denotes consumers' preference for the range performance of new energy vehicles, i.e., the degree to which consumers recognize the range performance of new energy vehicles. NEVs have intelligent components that distinguish them from CFVs and provide a better driving experience compared to CFVs, assuming that g denotes the level of intelligence of NEVs and ρ denotes consumers' smart preference for new energy vehicles, i.e., the degree of consumers' recognition of the intelligent performance of new energy vehicles, g∈[0,1],ρ∈[0,1].

Details are shown in the “Model description and assumptions “on line 400-412 of the revised version .

Point 4:Lack of Empirical Data: The article relies heavily on modeling and assumptions without presenting empirical data or real-world evidence to support its claims. Empirical validation of the model's assumptions and predictions would have added credibility to the study.

Response 4: Thank you for your comment. This is a very theoretical and practical question. This study constructs the density game model of new energy vehicles and conventional vehicles, taking into account both the maximum environmental capacity of the market and the game payment matrix. The maximum environmental capacity of the market in this study adopts the traditional fuel vehicle market sales volume selected from 1996-2018 as the research data to predict the maximum market size of the traditional fuel vehicles, and it has been supplemented with the model related parameter estimation process and results. The game payment matrix is based on the actual data related to the double integral policy, relevant literature, and the basic assumptions of the model, and the initial value of the model parameters are set. The research content of this paper is the evolutionary impact of consumer range preference and smart preference on the automobile market under the double integral policy since the implementation of the double integral policy in 2019, so the simulation start time of this paper is 2019, and the simulation study uses the data of 2020 as a test. The results of the simulation show that in 2020, the simulation of the new energy automobile market sales and the sales of conventional vehicle markets were 1,171,651,000 units and 2006,333,830,000 units, respectively, and their actual values were 1,246,000,000 units and 18,680,995,000 units, respectively, and the differences between the simulation results and the actual data are -6.35% and 6.89%, respectively, and the simulation results are basically consistent with the actual results. The additions are shown below.

Scenario simulation analysis

This study sets the parameters related to the density game model and makes initial value assumptions based on government and enterprise research data, related literature, and the basic assumptions of the model. The main parameter estimation and initial value settings are as follows.

Dynamic evolution of the independent development of CFVs

Logistic equations are widely used to analyze population growth and forecast market consumption demand. Without considering the competition for NEVs, the Dual Credit Policy, and consumption preference, the long-term development trend of CFVs shows the characteristics of the logistic growth curve [47]; the corresponding logistic equation is shown in Eq(23).

 y(t)=K_y/(1+(K_y/K_1 -1) e^(-r(t-t_0 ) ) ) (23)

where y denotes the market size of CFVs, r denotes the natural growth rate, and K_y denotes the maximum current market capacity of CFVs in the micro market.

In terms of parameter estimation, the parameter values in the logistic equation are also solved using historical statistics. NEVs and CFVs market sales volumes were not affected by the Dual Credit Policy before 2019. Moreover, the market sales volume data samples of NEVs are only accurate for the last 10 years; consequently, the maximum environmental capacity of NEVs aligns with the maximum environmental capacity of the CFV market. In this study, the market sales volume of conventional fuel passenger vehicles is selected as the research sample, excluding low-speed electric vehicles and agricultural vehicles. Therefore, this study selects the CFV market sales volume from 1996 to 2018 as the research data to predict the maximum market size of CFVs.

The research data of CFV sales volume from 1996 to 2018 are incorporated into equation (7), which is fitted using the nonlinear least squares method through MATLAB software. The fitting results are shown in Table 6 and Fig 3. Notably, the parameter equations are in good agreement with the actual data. According to the fitting results as shown in Table 6, the market sales of CFVs in China will gradually reach saturation around 2036, with a saturation value of about 26,663,000 units. Therefore, this study sets the maximum capacity of the market of CFVs, K_y, to 26.663 million units, and sets the maximum capacity of the NEV market, K_x, to 30 million units as the market for NEVs is expected to be more robust due to focus on sustainable developmen t [47].

Table 6. Results of equation parameter estimation

Parameter Estimated Value R^2

K_y 2666.3 0.993

Fig 3. Evolutionary trend of market share when CFVs are developed independently

Simulation background and initial value setting

This study follows Xuhui Wang et al.[48] in selecting a twofold basis for the range of parameter values. The first is the government and business research data and literature. In terms of policy research data, the Dual Credit Policy was officially implemented in 2019. The “Chinese passenger car manufacturers average fuel consumption and NEV credit accounting table” and “China Automotive Industry Yearbook,” set the initial value of 3.36 for a and 0.1 for δ. The average CAFC credits b initial value was 0.52. According to the data of the NEV points trading platform, the price of new energy credits in 2019–2022 was between 500–2088 yuan/points, and the initial value of the points price is set to be 0.05, p∈[0.05,0.2088]. In reference [22], let l=g=0.4,ρ=ϕ=0.5. In terms of corporate research data, the cost price of a CFV is approximately 60% of the selling price [35]. The second parameter is the principle of equilibrium. In the stability analysis of the density game model, in order to fulfil the realistic situation, it is necessary to satisfy that H_1-H_3-H_2+H_4<0and sgn⁡( H_1 H_4-H_2 H_3)=+1， and according to equations (8)–(13), the calculations are set to κ=1.2，C=0.1056，θ∈[0,1).

Accordingly, this study uses the 2020 data as a test. The simulation results show that the NEV and CFV market sales in 2020 are 1,171,651,000 and 2,006,383,300, respectively, while their actual values are 1,246,000,000 and 1,868,0995,000, respectively, resulting in a discrepancy of -6.35% and 6.89%, respectively. As the errors are less than 10% [49], the behavior and data described by the model are basically consistent with the actual system behavior and data, which proves the validity of the model and that it can be used for further research.

Details are shown in the “Dynamic evolution of the independent development of CFVs “and “ Simulation background and initial value setting”on line 803-891 of the revised version .

Point 5:Ambiguity in Findings: Despite the elaborate model and assumptions, the article's findings remain somewhat ambiguous. It discusses the impact of the Dual Credit Policy and consumer preferences on the NEV market but does not offer clear and actionable insights. The findings lack specificity and fail to provide guidance for policymakers or industry stakeholders.

Response 5: Thank you for your comment. We have rewritten our conclusions and added management recommendations.

Conclusion

In this study, we consider the competitive market environment and construct a competitive density game model of NEVs and CFVs considering the Dual Credit Policy and consumer preferences . Further, we explore the influence of consumers’ range preference and smart preference on the micro-production decision of automobile enterprises and the long-term evolution of the macro-automobile market under the Dual Credit Policy. 

(1) A low NEV credit price facilitates NEV market size growth, but this growth rate diminishes beyond a certain price threshold .When the NEV credit price is less than CNY 1204 per credit, increasing it can accelerate the growth of NEV market size to its saturation value, but it is difficult to increase the saturation value of NEV market size, and with the continuous growth of the credit price, this effect is weakened.

(2) The lower the consumer's range preference, the higher NEV credit price can accelerate the development of new energy vehicles to their saturation value. When consumers' smart preference is low, the increase in the NEV credit price is more significant enough to accelerate the development of NEVs to the saturation value of the market.

(3) Higher consumer preferences for both range and smart features, combined with increased NEV credit prices, can synergistically accelerate the speed of the NEV market to reach the saturation value and also raise the saturation value of the scale of NEVs. Higher consumer range preference combined with increased NEV credit prices has a more significant effect on the promotion of NEV market size than the combined effect of higher consumer smart preference and increased NEV credit prices. When consumers have the low smart preference and the high range preference for NEVs, compared to consumers have the high smart preference and the low range preference for NEVs, it is easier to increase the competitive advantage of NEVs, promote NEV market scale saturation value increase, and accelerate the time it takes NEVs to replace CFVs. 

The policy recommendations are as follows:

(1) The optimization of the Dual Credit Policy is necessary to combine the demand-side support environment, adjust the Dual Credit Policy standards and the overall trading mechanism. Further, it is important to give full play to the role of policy combinations, such as planning and supporting the development route of NEVs, providing support for strengthening the core technology of NEVs, introducing purchase subsidies, tax exemptions, and other related policies and measures. This can be synergistic with the Dual Credit Policy to help enterprises optimize according to consumer preferences, technical feasibility, and other available information. This can help make enterprises make scientific technology choices, focus on key challenges such as power battery range experience and solving consumers' mileage anxiety,promote consumer market-oriented purchase choices, improve product technology level, and promote the benign cycle of NEV research and development (R&D), production, and consumption. 

(2) Automakers should take technological innovation as a key element to improve the performance and battery capacity of the entire vehicle, reduce production costs, increase the price advantage, increase investment in the field of intelligent driving and other advantageous areas of NEVs, reduce the gap with conventional vehicles, improve the technical performance of NEVs to leverage the improvement in marketing efforts. 

Details are shown in the “Conclusion “on line 1375-1496 of the revised version.

Point 6:Disconnected Sections: The article appears disjointed at times, with sections discussing various aspects of the research without clear connections. It would benefit from better organization and a logical flow of information to aid comprehension.

Response 6: Thank you for your comment. Our article is structured as Section 1 Introduction discusses the research background and research significance of the article, Section 2 reviews the related literature, Section 3 describes the research methodology of the article, Section 4 proves and analyzes the game model, Section 5 performs numerical modeling and parametric analysis, and Section 6 summarizes the results of the article. We restructure the internal structure of Section 3 from the newly organized Section 3, and elaborate the chapter connection between “Competitive density game model stabilization point analysis” and “Competitive density game model stability analysis”. In the scenario simulation analysis in Section V, we added the parameter estimation of the maximum environmental capacity. Some parts of Section V appeared to be confusing due to the transfer of the pdf, and we revised the relevant content from new.

We have added the appropriate content at chapter transitions. The specific additions are as follows

From the stability analysis of the evolutionary game strategy, it can be seen that under the influence of the Dual Credit Policy, the evolutionary path of the decision-making of the automakers in the market regarding the type of automobiles to be produced mainly follows the process of the initial stage—evolutionary stage—ideal stage, and needs to be regulated by the Dual Credit Policy, in order to advance the system evolution. In the evolution process, there will be a section of the automakers that choose to produce NEVs, while the other section of automakers will continue to produce CFVs. These two types of automakers produce competition in the market. In order to study the specific impact of the Dual Credit Policy regulation means on the promotion of the production decision of the automakers, taking into account that the market is limited in resources, the automobile industry will be affected by the benefits of the market capacity constraints in the development of the automobile industry [36]; in fact, the party with higher benefits can more effectively utilize the ability of resources to achieve market diffusion, so further on in the competitive density game model for stability strategy solution.

Details are shown in the “Model analysis “on line 703-713 of the revised version.

Point 7:Language and Clarity: There are issues with language and clarity throughout the article. The writing can be convoluted and difficult to follow, making it challenging for readers to grasp the key points.

Response 7：Thank you for your comments. We have entrusted professionals to help improve the language of the paper. We performed the proofreading on the manuscript.

We deeply hope that our revised paper could meet your requirements. We have added line numbers in the revised manuscript. Thank you very much for your careful consideration.

Response to the Comments of Reviewer 2

Thank you for your instructive and valuable comments and suggestions on our manuscript “The Impact of Consumer Preferences on the Evolution of Competition in China's Automobile Market under the Dual Credit Policy - A Density Game Based Perspective”（PONE-D-23-29559）. We have studied all these comments, and have tried our best to revise the manuscript based thereon. The point-to-point responses to your comments are listed below. For ease of reading, the authors’ responses are listed in red color. Changes are also shown in red in the revised manuscript. 

Point 1:The academic writing in English leaves much to be desired, as the manuscript contains a high number of grammatical errors. The manuscript needs to be checked by the author in detail. 

Response 1：Thank you for your comments. We have entrusted professionals to help improve the language of the paper. We performed the proofreading on the manuscript.

Point 2:The formatting of the manuscript is confusing, with many serious errors such as figures and figure names not appearing on a single page, missing text, large gaps, etc. For example, in line 518 of the manuscript, the content after ‘In’ is missing. In line 521 of the manuscript, the text after ‘Combined with’ is missing. The manuscript needs to be verified by the author in detail. 

Response 2：Thank you for your comments. We have corrected the problem with lines 518 and 521 in our revised manuscript.

Details are shown in the revised manuscript on line 491-523 .

Point 3:The title of the manuscript uses 'Dual Integral Policy'. But in the paper as well as in the abstract, the reference is to 'Dual Credit Policy'. There is a difference between the two statements. It is recommended that the author study it in detail and unify it into a more relevant one. 

Response 3：Thank you for your comments. We have unified the policy statement as ”Dual Credit Policy”.

Details are shown in the revised manuscript .

Point 4:The manuscript is unclear in several places. For example, in line 183, the author mentions earlier that there are two automobile producers, each of which can produce two types of vehicles. Could the authors explain whether the intention here is to express that the basic benefits to consumers of the two types of vehicles produced by each producer (W1 or W2) want to be equal, or that the vehicles produced by both producers (W1 and W2) are equal?

Response 4：We are grateful for your comment, which has highlighted a need for greater clarity in our description of the strategic choices available to automakers 1 and 2. We have amended the text to explicitly state that both automakers have an identical set of strategic options, namely {Produce new energy vehicles, Produce conventional fuel vehicles}. This amendment rectifies the previous ambiguity and underscores the symmetrical nature of the decision-making framework within our model. The revised manuscript now clearly delineates these choices in the context of the competitive dynamics under investigation.

 Assumption 1: Automakers: the main body of the dual-credit policy assessment for passenger car manufacturers, considering the passenger car market there are two types of automakers, respectively known as automaker 1 and automaker 2, both engaged in automobile production activities in the automobile market, and production and sales balance; Automakers can only choose between producing new energy vehicles (NEVs) or producing conventional fuel vehicles (CFVs) in the market, and both of them have different maximum market capacities. automakers can choose from the same set of actions, i.e., {produce new energy vehicles, produce conventional fuel vehicles}. Automaker 1 chooses to produce new energy vehicles at a rate of m, while choosing to produce conventional fuel vehicles at a rate of 1-m, and Automaker 2 chooses to produce new energy vehicles at a rate of n, while choosing to produce conventional fuel vehicles at a rate of 1-n.The selling prices of new energy vehicles and conventional fuel vehicles are P_1 and P_2 respectively, the production costs are κC and C, where κ>1. The market demand for NEVs and CFVs respectively is D_1 and D_2, the profits are π_1 and π_2. 

Details are shown in the “Model description and assumptions “on line 188-199.

Point 5:In line 213 of the manuscript, the process of obtaining v1 through equations (1) and (2) represents the threshold of environmental awareness between no purchase behavior and the purchase of a NEV. Does it represent a problem with the conclusions derived from equations (1), (2), and (3), as well as the derivation of the later text, if it is written incorrectly here?

Response 5：Thank you for your comments. v_1indicates the threshold between consumers buying new energy vehicles and conventional fuel vehicles. There is no error in this calculation, and we have corrected the ambiguous statement here.The specific modifications are as follows：

 the utility of adopting NEVs u_1 :

 u_1=ν-p_1+ϕl+ρg (2)

 the utility of adopting CFVs u_2 : 

 u_2=θν-p_2+l (3)

 the utility of not buying any cars u_3:

 u_3=0 (4)

When u_1>u_2 and u_1>0, we can get ν>(l-ρg-ϕl+p_1-p_2)/(1-θ) and ν>p_1-ϕl-ρg, so that ν_1=(l-ρg-ϕl+p_1-p_2)/(1-θ), and when ν_1<ν<1, it is easy to know that the demand for new energy vehicles D_1=∫_(ν_1)^1▒f(ν) dν=1-(l-ρg-ϕl+p_1-p_2)/(1-θ). Whenu_1<u_2and u_2>0, we can get (-l-ηe_2+p_2)/θ<ν<(l-ρg-ϕl+p_1-p_2)/(1-θ), so that ν_2=(-l+p_2)/θ. When ν_2<ν<ν_1, the demand for conventional fuel vehicles D_2=∫_(ν_2)^(v_1)▒f(ν) dν=((l-p_2)-θ(ρg+ϕl-p_1))/θ(1-θ) . When u_2<u_3, we get 0<ν<(-l+p_2)/θ, consumers do not buy any cars .

Details are shown in the “Consumer utility function “on line 235-240.

Point 6:The manuscript proposes that ‘in the charging facilities in the construction of more backward areas, we should focus on the intelligent performance of new energy vehicles.’. Combined with real life, vehicle intelligence performance enhancement has little positive impact on consumers' choice of new energy vehicles if it is not convenient for consumers to charge.

Response 6：Thank you for your comments. This is a very good suggestion and we have rewritten the policy recommendations from scratch. The policy recommendations are set out below:

The policy recommendations are as follows:

(1) The optimization of the Dual Credit Policy is necessary to combine the demand-side support environment, adjust the Dual Credit Policy standards and the overall trading mechanism. Further, it is important to give full play to the role of policy combinations, such as planning and supporting the development route of NEVs, providing support for strengthening the core technology of NEVs, introducing purchase subsidies, tax exemptions, and other related policies and measures. This can be synergistic with the Dual Credit Policy to help enterprises optimize according to consumer preferences, technical feasibility, and other available information. This can help make enterprises make scientific technology choices, focus on key challenges such as power battery range experience and solving consumers' mileage anxiety,promote consumer market-oriented purchase choices, improve product technology level, and promote the benign cycle of NEV research and development (R&D), production, and consumption. 

(2) Automakers should take technological innovation as a key element to improve the performance and battery capacity of the entire vehicle, reduce production costs, increase the price advantage, increase investment in the field of intelligent driving and other advantageous areas of NEVs, reduce the gap with conventional vehicles, improve the technical performance of NEVs to leverage the improvement in marketing efforts. 

Details are shown in the “Conclusion “on line 545-559.

Point 7:The authors mainly consider the effects of changes in a single factor and do not consider the coupling between the factors, which is less relevant to the realities of life.

Response 7：We are grateful for your insightful comments. Your query indeed contributes substantially to the theoretical and practical richness of our work. In Section 5.4, titled "Consumer purchasing preferences and the evolution of the automobile market," we have expanded our discussion to encompass the effects of both single (range or smart preference) and combined (range and smart preferences) consumer behaviors on market dynamics at various price points. This analysis is grounded in comprehensive simulation studies, leading to a reconceptualization of our conclusions. Specific details are presented below:

Impact of consumer preferences on trends in automotive market evolution

As described by Shi [50]and Li[44], in order to achieve the sustained growth of new energy vehicle market size, policy intervention is necessary but not sufficient, because the market share growth of new energy vehicles is not only affected by macro policies, but also by consumer preferences, so in order to explore the impact of consumer preferences on the automobile market evolution, especially the NEV market evolution, under the Dual Credit Policy, we discuss how different consumer preferences affect the long-term evolution trend of the automobile market under different credit prices.

Impact of consumer range preference on automotive market evolution trends

Fig 5. shows the range preference of consumers for NEVs and the impact of the range performance of new energy vehicles on the evolutionary trend of the automobile market under different NEV credit prices, keeping the other parameters of the market competition system between NEVs and CFVs unchanged, and setting NEV credit price p=CNY 500 per credit,p=CNY 1204 per credit,p=CNY 2088 per credit, and consumers' range preference ϕ=0.1,ϕ=0.3,ϕ=0.5.

（a）p=CNY 500 per credit 

（b）p=CNY 1204 per credit 

（c）p=CNY 2088 per credit

Fig 5. Impact of consumer range preference on automotive market evolution trends

As shown in Fig 5.(a)-Fig 5.(c), when the NEV credit price is unchanged, the increase in consumers' range preference can enhance the saturation value of the development of the new energy vehicle market. This is because the combined effect of consumer range preference and the formation of NEV credit price makes NEVs more competitive in the market, leading to a competitive advantage to obtain the saturation value of a larger market size compared to competitors [51]. The lower the consumer's range preference, the higher NEV credit price can accelerate the development of new energy vehicles to their saturation value. Because, when consumer's range preference is low, the market acceptance of NEVs is low and the NEV credit price increases. This leads the government to utilize the credits deficit cost [52], which, in turn, prompts automakers to reduce the production of CFVs, focus on developing NEV range technology, subsidize the manufacturing cost of NEVs, and reduce the selling price of NEVs to increase consumer market demand for new energy vehicles.

Impact of consumer smart preference on automotive market evolution trends

Fig 6. shows the impact of consumer smart preference on the evolutionary trend of the automobile market under different NEV credit prices, keeping other parameters of the competitive system of the market for NEVs and CFVs unchanged, and setting NEV credit price p=CNY 500 per credit,p=CNY 1204 per credit, p=CNY 2088 per credit, and consumer smart preference ρ=0.1,ρ=0.3,ρ=0.5.

（a）p=CNY 500 per credit 

（b）p=CNY1204 per credit 

（c）p=CNY 1204 per credit

Fig 6. Impact of consumer smart preference on automotive market evolution trends 

As shown in Fig 6(a)- Fig 6 (c), when consumers in the market prioritize smart features, increasing the NEV credit price does not significantly influence the growth of NEV market size. When consumers' smart preference is low, the increase in the NEV credit price is more significant enough to accelerate the development of NEVs to the saturation value of the market. Because, when consumers' smart preference of NEVs is low, the market acceptance of NEVs is low. This leads the government to utilize the credits deficit cost [53], which, in turn, prompts automakers to increase the production of NEVs, focus on developing smart technology for NEVs to increase consumer market demand for NEVs.

Impact of dual consumer preferences on automobile market evolution trends

Fig 7. shows the impact of dual consumer preferences (range preference and smart preference) on the evolutionary trend of the automobile market under different NEV credit prices, keeping other parameters of the competitive system of the market for new energy vehicles and conventional vehicles constant, setting NEV credit price p=CNY 500 per credit, p=CNY 1204 per credit, p=CNY 2088 per credit, consumers' range preference ϕ=0.1,ϕ=0.3,ϕ=0.5, consumers' smart preference ρ=0.1,ρ=0.3,ρ=0.5.

（a）p=CNY 500 per credit 

（b）p=CNY1204 per credit 

（c）p=CNY 1204 per credit

Fig 7. Impact of dual consumer preferences (range and smart preference) on automobile market evolution trends

As shown in Fig 7.(a)–Fig 7.(c), when consumers' preferences for range and smart of NEVs in the market is low, the competitive advantage of CFVs is greater than that of NEVs, and it is difficult to increase the competitive advantage of NEVs by increasing the NEV credit price. As consumers' preference for the range and smart of NEVs increase, the increase in the NEV credit price can increase the time for NEVs to replace CFVs, indicating that consumers’ acceptance of NEVs is the prerequisite for their scalability [54]. Further, the Dual Credit Policy needs to be synergized with the consumers' willingness to buy NEVs. When consumers have the low smart preference and the high range preference for NEVs, compared to consumers have the high smart preference and the low range preference for NEVs, it is easier to increase the competitive advantage of NEVs, promote NEV market scale saturation value increase, and accelerate the time it takes NEVs to replace CFVs. It shows that consumers' higher range preference is more likely to increase the saturation value of NEV market size compared to consumers' improved smart preference. Accordingly, the higher range of NEVs is more likely to promote consumers' acceptance of NEVs than the smarter driving experience.

Details are shown in the “Impact of the Dual Credit Policy on the trend of automobile market evolution “on line 465-523.

We deeply hope that our revised paper could meet your requirements. We have added line numbers in the revised manuscript. Thank you very much for your careful consideration.

---

## [Decision Letter · Decision Letter 1]

4 Dec 2023

The Impact of Consumer Preferences on the Evolution of Competition in China's Automobile Market under the Dual Credit Policy - A Density Game Based Perspective

PONE-D-23-29559R1

Dear Dr. wu,

We’re pleased to inform you that your manuscript has been judged scientifically suitable for publication and will be formally accepted for publication once it meets all outstanding technical requirements.

Kind regards,

Grigorios L. Kyriakopoulos, 2 PhDs, 3 MSc, 2 MA, MEng, 2 BA, BSc

Academic Editor

PLOS ONE

Additional Editor Comments (optional):

Reviewers' comments:

Reviewer's Responses to Questions

**Comments to the Author**

1. If the authors have adequately addressed your comments raised in a previous round of review and you feel that this manuscript is now acceptable for publication, you may indicate that here to bypass the “Comments to the Author” section, enter your conflict of interest statement in the “Confidential to Editor” section, and submit your "Accept" recommendation.

Reviewer #2: All comments have been addressed

2. Is the manuscript technically sound, and do the data support the conclusions?

Reviewer #2: Yes

3. Has the statistical analysis been performed appropriately and rigorously? 

Reviewer #2: Yes

4. Have the authors made all data underlying the findings in their manuscript fully available?

Reviewer #2: Yes

5. Is the manuscript presented in an intelligible fashion and written in standard English?

Reviewer #2: Yes

6. Review Comments to the Author

Reviewer #2: The authors have adequately addressed my comments raised in a previous round of review and I feel that this manuscript is now acceptable for publication,I have no more comments.

7. PLOS authors have the option to publish the peer review history of their article (what does this mean?). If published, this will include your full peer review and any attached files.

Reviewer #2: **Yes: **Shibo Zhang

---

## [Editor Report · Acceptance letter]

20 Dec 2023

PONE-D-23-29559R1 

PLOS ONE

Dear Dr. wu, 

I'm pleased to inform you that your manuscript has been deemed suitable for publication in PLOS ONE. Congratulations! Your manuscript is now being handed over to our production team.

Kind regards, 

on behalf of

Dr. Grigorios L. Kyriakopoulos 

Academic Editor

PLOS ONE